# Understanding Cross-Domain Few-Shot Learning Based on Domain Similarity and Few-Shot Difficulty

**Jaehoon Oh**[*]
KAIST DS
Daejeon, South Korea
jhoon.oh@kaist.ac.kr

**Sungnyun Kim**[*]
KAIST AI
Seoul, South Korea
ksn4397@kaist.ac.kr

**Namgyu Ho**[*]
KAIST AI
Seoul, South Korea
itsnamgyu@kaist.ac.kr

**Jin-Hwa Kim**
NAVER AI Lab, SNU AIIS
Seongnam, South Korea
j1nhwa.kim@navercorp.com

**Hwanjun Song**[†]
NAVER AI Lab
Seongnam, South Korea
hwanjun.song@navercorp.com

**Se-Young Yun**[†]
KAIST AI
Seoul, South Korea
yunseyoung@kaist.ac.kr

## Abstract

Cross-domain few-shot learning (CD-FSL) has drawn increasing attention for handling large differences between the source and target domains–an important concern in real-world scenarios. To overcome these large differences, recent works have considered exploiting small-scale unlabeled data from the target domain during the pre-training stage. This data enables self-supervised pre-training on the target domain, in addition to supervised pre-training on the source domain. In this paper, we empirically investigate which pre-training is preferred based on *domain similarity* and *few-shot difficulty* of the target domain. We discover that the performance gain of self-supervised pre-training over supervised pre-training becomes large when the target domain is dissimilar to the source domain, or the target domain itself has low few-shot difficulty. We further design two pre-training schemes, mixed-supervised and two-stage learning, that improve performance. In this light, we present six findings for CD-FSL, which are supported by extensive experiments and analyses on three source and eight target benchmark datasets with varying levels of domain similarity and few-shot difficulty. Our code is available at https://github.com/sungnyun/understanding-cdfsl.

## 1 Introduction

Few-shot learning (FSL) is a machine learning paradigm to learn novel classes from *few* examples with supervised information [66, 69]. Unlike standard supervised learning, a model is pre-trained on the source dataset consisting of *base* classes and then transferred into the target dataset consisting of *novel* classes with few examples, where base and novel classes are disjoint but share similar data domains. However, this underlying assumption is not applicable to real-world scenarios because source (base classes) and target (novel classes) domains are different in general. This leads to poor generalization performance because of the change in feature and label distributions, posing a new challenge in FSL [24, 64].

In this regard, *cross-domain few-shot learning* (CD-FSL) is gaining immense attention with the BSCD-FSL (Broader Study of CD-FSL) benchmark [24], which enables us to evaluate real-world few-shot learning tasks. The BSCD-FSL benchmark is a collection of four different datasets with varying

---

[*]Equal contribution.
[†]Corresponding authors.

36th Conference on Neural Information Processing Systems (NeurIPS 2022).

levels of domain similarity to large-scale natural image collections, such as ImageNet [11]. Although there are two possible directions for FSL, meta-learning [17, 35, 64] and transfer learning [4, 12, 62], transfer learning has been reported to have higher performance than meta-learning approaches in cross-domain scenarios. Therefore, following the transfer learning pipeline, recent studies for CD-FSL [47, 30] have mainly focused on improving the pre-training phase before fine-tuning on the target labeled data with novel classes.

To address the challenge of different domains, there have been recent efforts to leverage *unlabeled* examples from the target domain as auxiliary data for pre-training, in addition to labeled examples from the source domain. For example, along with the supervised cross-entropy loss, STARTUP [47] and Dynamic Distillation [30] incorporate distillation loss and FixMatch-like loss for self-supervision, respectively. In other words, they develop sophisticated pre-training approaches that can leverage source and target data together. However, the basic pre-training schemes, supervised learning (SL) on the source domain and self-supervised learning (SSL) on the target domain, have not been thoroughly studied with respect to their pros and cons in CD-FSL.

In this paper, we establish an *empirical understanding* of the effectiveness of SL and SSL for a better pre-training process in CD-FSL. To this end, we begin by scrutinizing an opposing finding of the previous works [47, 30]. We discover that readily available SSL methods, *e.g.*, SimCLR [3], can outperform the standard SL method for pre-training, even when the amount of unlabeled target data for SSL is much smaller than that of labeled source data for SL (see Section 4).

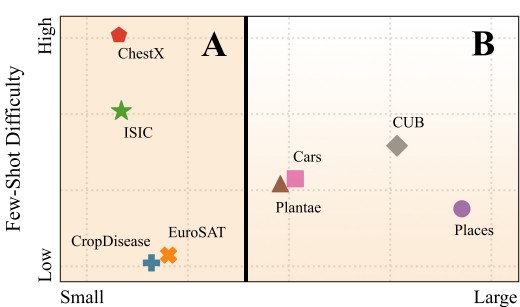

Figure 1: Our insights on the pre-training approaches. (A) SSL is preferred for all datasets with small domain similarity. (B) SL is preferred for high-difficulty datasets with large domain similarity. The formal definitions of similarity and difficulty are explained in Section 3.2.

Next, we investigate why the CD-FSL performance depends on different pre-training schemes using the two properties: *domain similarity* and *few-shot difficulty*. **Domain Similarity** is the similarity between the source and target domains, which is known to affect the transferability of the source domain features into the target domain [10, 36]. However, we find it insufficient to identify the effectiveness of SL and SSL based on domain similarity alone. To solve this conundrum, we propose **Few-Shot Difficulty** as a measure of the inherent hardness of a dataset, based on the upper bound of empirical FSL performance. By grounding our analysis on these two metrics, we discover coherent insights on CD-FSL pre-training schemes, depicted in Figure 1. Our analyses point to two conclusions: (A) When domain similarity is small, SSL is preferred due to the limited transferability of source information. On the other hand, (B) SL is preferred when domain similarity is large and few-shot difficulty is high, because supervision from the source dataset achieves stronger performance compared to self-supervision on difficult target data (see Section 5).

Finally, to investigate whether SL and SSL can synergize, we design a joint learning scheme using both SL and SSL, coined as *mixed-supervised learning* (MSL). It is observed that SL and SSL can synergize when they have similar performances. Furthermore, we extend our analysis to a *two-stage* pre-training scheme, motivated by recent works on CD-FSL [47, 30]. We observe that this generally improves performance because the SL pre-trained model provides a good initialization for the second phase of pre-training (see Section 6).

## 2 Related Work

### 2.1 Few-Shot Learning (FSL)

FSL has been mainly studied in the literature based on two approaches, meta-learning and transfer learning. In the meta-learning approach, a model is trained on the meta-train set (*i.e.*, source data) in an episodic manner, mimicking the evaluation procedure, such that fast adaptation is possible on the meta-test set (*i.e.*, few-shot target data). This family of approaches include learning a good

initialization [38, 17, 18, 48, 43], learning a metric space [66, 54, 58], and learning an update rule [50, 1, 19]. By contrast, in the transfer-based approach [4, 12, 62], a model is pre-trained on the source dataset following the general supervised learning procedure in a mini-batch manner, and subsequently fine-tuned on the target dataset for evaluation.

## 2.2 Cross-Domain Few-Shot Learning (CD-FSL)

CD-FSL has addressed a more challenging and realistic scenario where the source and target domains are dissimilar [24, 64]. Such a cross-domain setting makes it difficult to transfer source information into the target domain owing to large domain differences [44, 36, 40, 73]. In general, the most recent methods have been developed on top of the fine-tuning paradigm because this paradigm outperforms the traditional meta-learning approach such as FWT [24]. STARTUP [47] and Dynamic Distillation [30] are the two representative algorithms, and they suggested using small-scale unlabeled data from the target domain in pre-training such that a pre-trained model can be well-adaptable for the target domain. Specifically, both algorithms first train a teacher network with cross-entropy loss on labeled source data. Then, STARTUP trains the student network with cross-entropy loss on the source data together with two unsupervised losses on the target data: distillation loss [28] and self-supervised loss (*i.e.*, SimCLR [3]). Dynamic Distillation trains the student network with cross-entropy loss on labeled source data and KL loss based on FixMatch [55] on unlabeled target data.

## 2.3 Self-Supervised Learning (SSL)

SSL has attracted attention as a method of learning useful representations from unlabeled data [14, 13, 72, 46, 42]. When this field first emerged, hand-crafted pretext tasks, such as solving jigsaw puzzles [41] and predicting rotations [20], were designed and utilized for training. In recent times, there has been an effort to use contrastive loss, which enhances representation learning based on augmentation and negative samples [3, 61, 26, 2]. This contrastive loss encourages the alignment of positive pairs and uniformity of data distribution on the hypersphere [67]. This improves the transferability of a model by encouraging it to contain lower-level semantics compared to supervised approaches [31]. However, this advantage is conditional on the availability of numerous negative samples. To alleviate such constraint, non-contrastive approaches that do not use negative samples have been proposed [23, 5, 71, 63]. In our empirical study, we use two contrastive approaches, SimCLR [3] and MoCo [26], and two non-contrastive approaches, BYOL [23] and SimSiam [5]. The details of each algorithm are described in Appendix A.

For the completeness of our survey, we include prior works that address SSL for cross-domain and/or few-shot learning. Kim et al. [32] addressed self-supervised pre-training under label-shared cross-domain, while our setting does not share the label space between domains. Ericsson et al. [16] observed that SSL on the source data improves performance on the BSCD-FSL dataset. However, domain-specific SSL (*i.e.*, SSL on target data) was not addressed. Cole et al. [8] showed that adding data from different domains can lead to performance degradation when data is numerous. Phoo and Hariharan [47] and Islam et al. [30] argued that plain SSL methods struggle to outperform SL for CD-FSL. We investigate domain-specific SSL and demonstrate its superiority, which opposes the finding from previous studies.

## 3 Overview

We clarify the scope of our empirical study, propose formal definitions of domain similarity and few-shot difficulty, and describe experimental configurations. Table 1 summarizes the notations used in this paper.

### 3.1 Scope of the Empirical Study

Our objective is to learn a feature extractor $f$ on base classes $\mathcal{C}_B$ in source data $\mathcal{D}_B$, which can extract informative representations for novel classes

Table 1: Summary of the notations.

| Notation | Description |
|---|---|
| $\mathcal{D}_B, \mathcal{D}_N$ | Source and target datasets, $\mathcal{D}_B \cap \mathcal{D}_N = \emptyset$ |
| $\mathcal{C}_B, \mathcal{C}_N$ | Base classes for $\mathcal{D}_B$ and novel classes for $\mathcal{D}_N$ |
| $\mathcal{D}_U \ (\subset \mathcal{D}_N)$ | Unlabeled target data for SSL |
| $\mathcal{D}_L \ (\subset \mathcal{D}_N)$ | Labeled target data for evaluation, $\mathcal{D}_U \cap \mathcal{D}_L = \emptyset$ |
| $n, k$ | # classes and examples for $n$-way $k$-shot |
| $\mathcal{D}_S \ (\subset \mathcal{D}_L)$ | A support set with size $nk$ for fine-tuning |
| $\mathcal{D}_Q \ (\subset \mathcal{D}_L)$ | A query set for evaluation, $\mathcal{D}_S \cap \mathcal{D}_Q = \emptyset$ |
| $f$ | A feature extractor (backbone network) |
| $h_{\mathsf{sl}}$ | A classification head for SL during pre-training |
| $h_{\mathsf{ssl}}$ | A projection head for SSL during pre-training |
| $g$ | A classification head during fine-tuning |

$\mathcal{C}_N$ in target data $\mathcal{D}_N$. Typically, a classifier $g$ is fine-tuned and the model $g \circ f$ is evaluated using labeled target examples $\mathcal{D}_L$ $(\subset \mathcal{D}_N)$ after pre-training $f$ on the source data $\mathcal{D}_B$ under the condition that the base classes are largely different from the novel classes.

Following the recent literature [47, 30], we further assume that additional unlabeled data $\mathcal{D}_U$ $(\subset \mathcal{D}_N)$ is available in the pre-training phase. We follow the split strategy used in Phoo and Hariharan [47], where 20% of the target data $\mathcal{D}_N$ is used as the unlabeled data $\mathcal{D}_U$ for pre-training. Note that the size of the unlabeled portion is very small (*e.g.*, only a few thousand examples) compared to large-scale datasets typically considered for self-supervised learning. In this problem setup, the pre-training phase of CD-FSL can be carried out based on *three* learning strategies:

- **Supervised Learning**: Let $f$ and $h_{\mathsf{sl}}$ be the feature extractor and linear classifier for the base classes $\mathcal{C}_B$, respectively. Then, a model $h_{\mathsf{sl}} \circ f$ is pre-trained only for the labeled source data $\mathcal{D}_B$ by minimizing the standard cross-entropy loss $\ell_{\mathsf{ce}}$ in a mini-batch manner.[3]

$$\mathcal{L}_{\mathsf{sl}}(f, h_{\mathsf{sl}}; \mathcal{D}_B) = \frac{1}{|\mathcal{D}_B|} \sum_{(x,y) \in \mathcal{D}_B} \ell_{\mathsf{ce}}(h_{\mathsf{sl}} \circ f(x), y). \tag{1}$$

- **Self-Supervised Learning**: Let $h_{\mathsf{ssl}}$ be the projection head. Then, a model $h_{\mathsf{ssl}} \circ f$ is pre-trained only for the unlabeled target data $\mathcal{D}_U$, which is much smaller than the labeled source data, by minimizing (non-)contrastive self-supervised loss $\ell_{\mathsf{self}}$ (*e.g.*, NT-Xent),[1]

$$\mathcal{L}_{\mathsf{ssl}}(f, h_{\mathsf{ssl}}; \mathcal{D}_U) = \frac{1}{2|\mathcal{D}_U|} \sum_{x \in \mathcal{D}_U} \left[ \ell_{\mathsf{self}}(z_1; z_2; \{z^-\}) + \ell_{\mathsf{self}}(z_2; z_1; \{z^-\}) \right] \tag{2}$$

$$\text{where } z_i = h_{\mathsf{ssl}} \circ f(A_i(x)),$$

and $A_i(x)$ is the $i$-th augmentation of the same input $x$. This training loss forces $z_1$ to be similar to $z_2$ and dissimilar to the set of negative features $\{z^-\}$. In addition, there are non-contrastive SSL methods that do not rely on negative examples, *i.e.*, $\{z^-\} = \emptyset$. We provide a more detailed explanation of SSL losses, including multiple (non-)constrastive approaches in Appendix A.

- **Mixed-Supervised Learning**: MSL exploits labeled as well as unlabeled data from different domains simultaneously. MSL can be intuitively formulated by minimizing the interpolation of their losses in Eqs. (1) and (2),

$$\mathcal{L}_{\mathsf{msl}}(f, h_{\mathsf{sl}}, h_{\mathsf{ssl}}; \mathcal{D}_B, \mathcal{D}_U) = (1 - \gamma) \cdot \mathcal{L}_{\mathsf{sl}}(f, h_{\mathsf{sl}}; \mathcal{D}_B) + \gamma \cdot \mathcal{L}_{\mathsf{ssl}}(f, h_{\mathsf{ssl}}; \mathcal{D}_U), \tag{3}$$

where $0 < \gamma < 1$ and the feature extractor $f$ is hard-shared and trained through SL and SSL losses with a balancing hyperparameter $\gamma$. This can be a generalization of STARTUP and Dynamic Distillation, which use Eq. (3) in the second pre-training phase with a moderate modification after the typical pre-training phase using SL.

Our analysis focuses on pre-training and fine-tuning schemes due to the superiority of transfer-based methods over typical FSL algorithms such as MAML [17], which is shown in [24]. Based on the three learning strategies above, we conduct an empirical study to gain an in-depth understanding of their effectiveness in the pre-training phase, providing deep insight into the following questions:

1. Which is more effective for pre-training, using only SL or SSL? ▷ Section 4

2. How to apply domain similarity and few-shot difficulty to identify the more effective pre-training scheme between SL and SSL, for CD-FSL? ▷ Section 5

3. Can MSL, a combination of SL and SSL, as well as a two-stage scheme improve performance? ▷ Section 6

## 3.2 Domain Similarity and Few-Shot Difficulty

We present a procedure for estimating the two metrics on datasets, which are used to analyze the pre-training schemes. First, we use *domain similarity* introduced in [10], which is based on Earth Mover's Distance (EMD [52]) because the distance between the two domains can be considered as

---

[3]The batch loss on the entire data is used for ease of exposition.

the cost of *moving* images from one domain to the other in the transfer learning context [10, 36]. Further details on this metric, *e.g.*, advantages of EMD, are explained in Appendix D.

We can easily compute EMD using the retrieved sample representations.[4] We create the prototype vector $\mathbf{p}_i$, which is an averaged representation for all examples belonging to class $i$. Next, let $i \in \mathcal{C}_B$ and $j \in \mathcal{C}_N$ be a class in base (source) and novel (target) classes, respectively. Then, the domain similarity between the source and target data is formulated as

$$\text{Sim}(\mathcal{D}_B, \mathcal{D}_N) = \exp\big(-\alpha \, \text{EMD}(\mathcal{D}_B, \mathcal{D}_N)\big) \quad \text{where} \;\; \text{EMD}(\mathcal{D}_B, \mathcal{D}_N) = \frac{\sum_{i \in \mathcal{C}_B, j \in \mathcal{C}_N} f_{i,j} \, d_{i,j}}{\sum_{i \in \mathcal{C}_B, j \in \mathcal{C}_N} f_{i,j}}$$

$$\text{subject to} \;\; f_{i,j} \geq 0, \;\; \sum_{i \in \mathcal{C}_B, j \in \mathcal{C}_N} f_{i,j} = 1, \;\; \sum_{j \in \mathcal{C}_N} f_{i,j} \leq \frac{|\mathcal{D}_B[i]|}{|\mathcal{D}_B|}, \;\; \sum_{i \in \mathcal{C}_B} f_{i,j} \leq \frac{|\mathcal{D}_N[j]|}{|\mathcal{D}_N|}, \quad (4)$$

where $d_{i,j} = ||\mathbf{p}_i - \mathbf{p}_j||_2$; $f_{i,j}$ is the optimal flow between $\mathbf{p}_i$ and $\mathbf{p}_j$ subject to the constraints for EMD; $\mathcal{D}[i]$ returns all examples of the specified class $i$ in $\mathcal{D}$; and $\alpha$ is typically set to 0.01 [10]. Namely, EMD can be interpreted as the weighted distance of all combinations between the base and novel classes. The larger similarity indicates that source and target data share similar domains.

Next, we propose *few-shot difficulty*, which quantifies the difficulty of a dataset based on the empirical upper bound of few-shot performance in our problem setup, regardless of its relationship to the source dataset. To capture the upper bound of FSL performance, we use 20% of the target dataset as labeled data to pre-train the model in a supervised manner. Then, the pre-trained model is evaluated on the remaining unseen target data for the 5-way $k$-shot classification task.[5] As the generalization capability indicates the hardness [56], the classification accuracy for unseen data is used and converted into the few-shot difficulty using an exponential function with a hyperparameter $\beta$ (the default value is 0.01),

$$\text{Diff}(\mathcal{D}, k) = \exp(-\beta \, \text{Acc}(\mathcal{D}, k)), \quad (5)$$

where $\text{Acc}(\mathcal{D}, k)$ returns the average of 5-way $k$-shot classification accuracy over 600 episodes for the given data $\mathcal{D}$. Note that in our paper, $k$ is set to 5, but the order of difficulty is the same regardless of $k$. High few-shot difficulty implies that the achievable accuracy is low even when there is no domain difference between pre-training and evaluation.

### 3.3 Experimental Configurations

**Cross-Domain Datasets.** We use ImageNet, tieredImageNet, and miniImageNet as source datasets for generality. Regarding the target domain, we prepare eight datasets with varying domain similarity and few-shot difficulty; domain similarity is computed based on both the source and target datasets, while few-shot difficulty is computed based on the target dataset. To summarize their order in Figure 1, ***domain similarity** to ImageNet: Places > CUB > Cars > Plantae > EuroSAT > CropDisease > ISIC > ChestX*, and **few-shot difficulty**: *ChestX > ISIC > CUB > Cars > Plantae > Places > EuroSAT > CropDisease*. For instance, Places data has the largest domain similarity to ImageNet, while ChestX has the highest few-shot difficulty. Appendix B provides the details of each dataset. The detailed values for domain similarity and few-shot difficulty are reported in Appendix D and E, respectively. These are visualized in Appendix F. We also provide the results on the case when source and target domains are the same, *i.e.*, the standard FSL setting, in Appendix O.

**Evaluation Pipeline.** We follow the standard evaluation pipeline of CD-FSL [24]. The evaluation process is performed in an episodic manner, where each episode represents a distinct few-shot task. Each episode is comprised of a support set $\mathcal{D}_S$ and a query set $\mathcal{D}_Q$, which are sampled from the entire labeled target data $\mathcal{D}_L$. The support set $\mathcal{D}_S$ and query set $\mathcal{D}_Q$ consist of $n$ classes that are randomly selected among the entire set of novel classes $\mathcal{C}_N$. For the $n$-way $k$-shot setting, $k$ examples are randomly drawn from each class for the support set $\mathcal{D}_S$, while $k_q$ (typically 15) examples for the query set $\mathcal{D}_Q$. Thus, the support and query set are defined as,

$$\mathcal{D}_S = \{(x_i^s, y_i^s)\}_{i=1}^{n \times k} \;\; \text{and} \;\; \mathcal{D}_Q = \{(x_i^q, y_i^q)\}_{i=1}^{n \times k_q}. \quad (6)$$

---

[4]To extract the representation of images, we follow Li et al. [36] by using a large model trained on a large-scale dataset, ResNet101 pre-trained on ImageNet. Note that Cui et al. [10] used JFT dataset [57], which is not released for public use. Furthermore, we measure domain similarity using different feature extractors, described in Table 6 of Appendix D. Our analysis is consistent regardless of the feature extractor used.

[5]We use a few-shot learning task instead of classification on the entire data, preventing the performance from being distorted by other factors, such as data imbalance and the number of classes.

Table 2: 5-way $k$-shot CD-FSL performance (%) of the models pre-trained by SL and SSL. We report the average accuracy and its 95% confidence interval over 600 few-shot episodes. B and S indicate base and strong augmentation, respectively. The best accuracy is marked in bold for each backbone.

| Source Data | Pre-train Scheme | Method | Aug. | EuroSAT k=1 | EuroSAT k=5 | CropDisease k=1 | CropDisease k=5 | ISIC k=1 | ISIC k=5 | ChestX k=1 | ChestX k=5 |
|---|---|---|---|---|---|---|---|---|---|---|---|
| ImageNet | SL | Default | B | 66.14±.83 | 84.73±.51 | 74.18±.82 | 92.81±.45 | 31.11±.55 | 44.10±.58 | 22.48±.39 | 25.51±.44 |
| tiered ImageNet | SL | Default | B | 61.81±.88 | 79.87±.67 | 66.82±.90 | 87.19±.59 | 30.35±.60 | 41.67±.55 | 22.34±.38 | 25.08±.45 |
| | | | S | 60.07±.88 | 79.95±.66 | 65.70±.94 | 86.34±.60 | 29.75±.56 | 40.60±.58 | 22.11±.42 | 25.20±.41 |
| - | SSL | SimCLR | B | 70.37±.86 | 87.80±.46 | 90.94±.69 | 97.44±.29 | 34.13±.69 | 44.37±.66 | 21.41±.41 | 25.05±.42 |
| | | | S | **84.30**±.73 | **94.12**±.32 | **91.00**±.76 | **97.46**±.34 | **36.39**±.66 | 47.85±.65 | 21.55±.41 | 25.26±.44 |
| | | MoCo | B | 51.21±.93 | 68.19±.74 | 70.22±.95 | 87.11±.60 | 27.79±.53 | 36.60±.59 | 21.44±.43 | 24.28±.43 |
| | | | S | 69.11±.98 | 81.01±.73 | 80.08±.97 | 92.48±.52 | 29.54±.59 | 39.28±.58 | 21.74±.42 | 24.58±.44 |
| | | BYOL | B | 60.98±.91 | 84.88±.56 | 81.58±.78 | 96.82±.27 | 35.31±.64 | **49.26**±.64 | 22.65±.42 | **28.80**±.49 |
| | | | S | 66.16±.86 | 87.83±.48 | 85.77±.73 | 96.93±.30 | 34.53±.62 | 47.59±.63 | **22.75**±.41 | 28.36±.46 |
| | | SimSiam | B | 44.06±.86 | 61.03±.72 | 75.36±.82 | 92.31±.44 | 26.99±.52 | 35.68±.52 | 22.02±.41 | 26.06±.46 |
| | | | S | 70.80±.88 | 85.10±.57 | 84.72±.80 | 96.05±.36 | 30.17±.56 | 39.51±.55 | 22.17±.40 | 26.56±.46 |

(a) ResNet18 is used as a backbone.

| Source Data | Pre-train Scheme | Method | Aug. | EuroSAT k=1 | EuroSAT k=5 | CropDisease k=1 | CropDisease k=5 | ISIC k=1 | ISIC k=5 | ChestX k=1 | ChestX k=5 |
|---|---|---|---|---|---|---|---|---|---|---|---|
| mini ImageNet | SL | Default | B | 64.03±.91 | 82.72±.59 | 73.38±.87 | 91.53±.49 | 30.68±.58 | 41.77±.59 | 22.64±.40 | 26.26±.45 |
| | | | S | 65.03±.88 | 84.00±.56 | 72.82±.87 | 91.32±.49 | 29.91±.54 | 40.84±.56 | 22.88±.42 | 27.01±.44 |
| - | SSL | SimCLR | B | 66.77±.84 | 86.39±.48 | 89.33±.66 | 96.82±.32 | 33.32±.63 | 44.50±.64 | 22.26±.42 | 24.34±.42 |
| | | | S | **79.50**±.78 | **92.36**±.37 | **89.49**±.74 | **97.24**±.33 | **34.90**±.64 | **46.76**±.61 | 21.97±.41 | 25.62±.43 |
| | | MoCo | B | 48.70±.92 | 66.85±.72 | 68.77±.92 | 87.67±.57 | 27.76±.54 | 38.03±.57 | 21.55±.42 | 24.48±.44 |
| | | | S | 76.20±.89 | 89.54±.46 | 80.19±.99 | 93.41±.53 | 30.20±.55 | 41.14±.57 | 21.64±.40 | 24.49±.43 |
| | | BYOL | B | 61.18±.82 | 83.11±.57 | 80.50±.75 | 94.85±.35 | 33.02±.62 | 46.72±.65 | 22.90±.41 | 27.40±.47 |
| | | | S | 66.45±.80 | 86.55±.50 | 80.10±.76 | 94.53±.41 | 33.50±.59 | 45.99±.63 | 23.11±.42 | 27.71±.44 |
| | | SimSiam | B | 44.57±.82 | 63.67±.67 | 82.83±.73 | 95.37±.34 | 30.74±.60 | 41.28±.62 | 22.76±.42 | 27.50±.47 |
| | | | S | 71.66±.88 | 85.21±.59 | 81.25±.77 | 95.13±.37 | 31.80±.59 | 41.44±.59 | **23.22**±.41 | **27.83**±.46 |

(b) ResNet10 is used as a backbone.

For evaluation, a classifier $g$ is fine-tuned on the support set $\mathcal{D}_S$, using features extracted from the fixed pre-trained backbone $f$. Note that $g$ is for the evaluation purpose different from $h_{sl}$ and $h_{ssl}$ for pre-training. The fine-tuned model $g \circ f$ is then tested on the query set $\mathcal{D}_Q$. We set $n = 5$ and $k = \{1, 5\}$, and the accuracy is averaged over 600 episodes following convention [24, 47].

**Implementation.** We use different backbone networks depending on the source data. For ImageNet and tieredImageNet, ResNet18 is used as the backbone, while ResNet10 is used for miniImageNet. For ResNet18 pre-trained on ImageNet with SL, we use the model provided by PyTorch [45] repository. This setup is exactly the same for all pre-training schemes. Additional details on the training setup are provided in Appendix C.

## 4 Supervised Learning on Source vs. Self-Supervised Learning on Target

We begin by investigating the superiority of SSL on the target dataset over SL on the source dataset for pre-training. We compare the CD-FSL performance of pre-trained models using four representative (widely cited) SSL methods (SimCLR [3], MoCo [26], BYOL [23], and SimSiam [5]) with that of an SL method (Default) in Table 2. Four different domain datasets from the BSCD-FSL benchmarks (EuroSAT, CropDisease, ISIC, and ChestX) are used as target data. Table 2 provides empirical evidence of the findings in this section. Recent literature has reported that SSL pre-training does *not* work better than SL for the CD-FSL task because of insufficient unlabeled examples in the target domain [47, 30]. However, our observation contradicts this previous finding.

OBSERVATION 4.1. *SSL on the target domain can achieve remarkably higher performance over SL on the labeled source domain, even with small-scale (i.e., a few thousand) unlabeled target data.*

EVIDENCE. SSL methods are observed to outperform SL in most cases, even though SSL does not leverage source data for pre-training. In particular, SSL methods show much higher performance compared to the model pre-trained on the entire ImageNet dataset, which has more than 1.2M training examples. This leads to the conclusion that SSL on the target domain can be better than SL on the source domain for CD-FSL pre-training. In other words, unlabeled target data available at the pre-training phase is worth more than labeled source data, even if the unlabeled target data is much smaller (*e.g.*, 8k examples for CropDisease) than the labeled source data. In Appendix G, we show that SSL can outperform SL using even smaller portions of unlabeled target data.

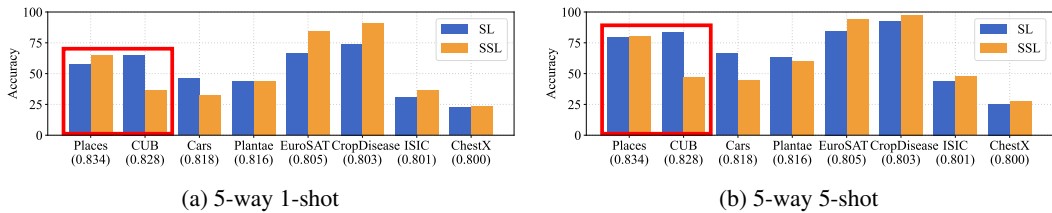

|  (a) 5-way 1-shot | (b) 5-way 5-shot |

Figure 2: 5-way $k$-shot CD-FSL performance (%) of SL and SSL according to domain similarity (values in x-axis), with ImageNet source data. The red box shows that SL outperforms SSL in the second largest domain similarity, while SSL outperforms SL in the largest domain similarity.

OBSERVATION 4.2. *SSL achieves significant performance gains with strong data augmentation.*

EVIDENCE. In addition, the results in Table 2 provide the performance sensitivity to data augmentation. For this study, two types of augmentation are used: (1) base augmentation from [3], which consists of random resized crop, color jitter, horizontal flip, and normalization, and (2) strong augmentation from [30], which adds Gaussian blur and random gray scale to the base (see the detail of the augmentations in Appendix C). With strong augmentation, SSL methods exhibit significant performance gains of up to 27.50%p compared to base augmentation, *i.e.*, MoCo on EuroSAT in Table 2(b). However, SL does not benefit from strong augmentation as SSL does. This has also been observed in the literature [3]. Therefore, the performance of SSL can be further improved for CD-FSL if more suitable augmentation is applied. Based on this observation, we use strong augmentation for SSL as the default setup in the rest of our paper.

Meanwhile, the superiority among SSL algorithms varies with target dataset. In Table 2, we observe that SimCLR performs best in EuroSAT and CropDisease, while in ISIC, SimCLR and BYOL both perform well. For ChestX, BYOL and SimSiam show good performance. The SSL methods can be categorized into two groups: contrastive (SimCLR and MoCo) and non-contrastive (BYOL and SimSiam). For the rest of our paper, we focus our analysis on SimCLR and BYOL, which are representative methods from each group with robust performance. The results for other target datasets are presented in Appendix H.

## 5 Closer Look at Domain Similarity and Few-Shot Difficulty

We investigate why the CD-FSL performance depends on different pre-training schemes, *i.e.,* SL or SSL, based on the two metrics: domain similarity and few-shot difficulty in Eqs. (4) and (5). We analyze the relationship between few-shot performance and the two metrics on various target datasets and provide insights for developing a more effective pre-training approach.

Including BSCD-FSL, we consider four additional datasets from different domains: Places, CUB, Cars, and Plantae. Note that these additional datasets are known to be more similar to ImageNet than the BSCD-FSL datasets are [16], and our estimated similarity shows the same trend. We mainly use ImageNet as the source dataset to make our analysis more reliable. We analyze their domain similarity and few-shot difficulty and display them in Figure 1, where ImageNet is used as source data for domain similarity. In this section, to select the SSL method for each dataset, we use SimCLR for all datasets except ChestX, where BYOL is used, based on the performance observed in Section 4.

### 5.1 Domain Similarity

Figure 2 shows the CD-FSL performance of the pre-trained models using SL and SSL for eight target datasets with varying domain similarity, where all the datasets are sorted by domain similarity. A common belief about domain similarity is that, as domain similarity increases, it is more beneficial for pre-training to use a large amount of labeled source data [10, 36, 16]. Our analysis shows that this belief is partially true.

OBSERVATION 5.1. *SL does not consistently benefit from large domain similarity.*

EVIDENCE. For the aforementioned belief to be true, the performance gain of SL over SSL should be greater as domain similarity increases. However, although SL outperforms SSL in the CUB dataset

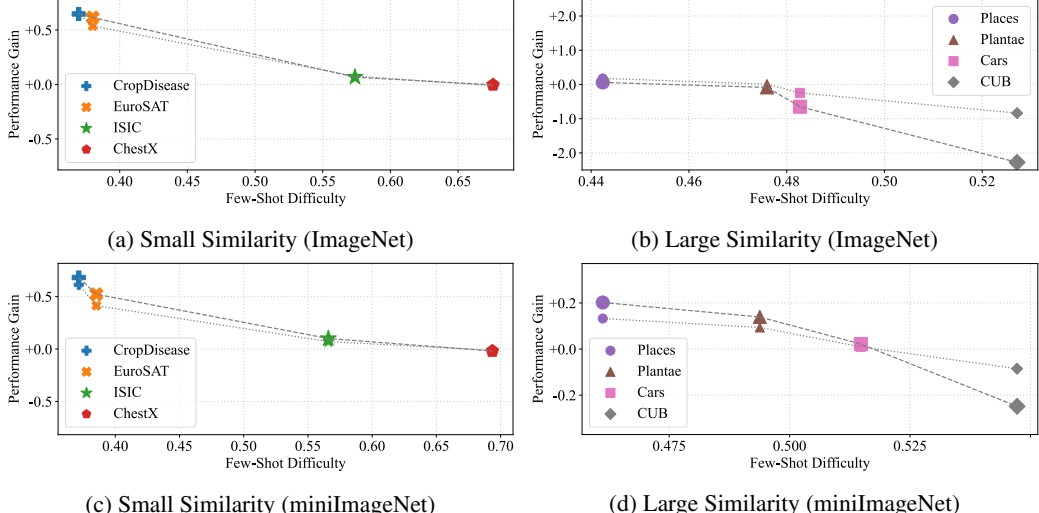

(a) Small Similarity (ImageNet)    (b) Large Similarity (ImageNet)

(c) Small Similarity (miniImageNet)    (d) Large Similarity (miniImageNet)

Figure 3: 5-way $k$-shot performance gain of SSL over SL for the two dataset groups according to the few-shot difficulty (small markers: $k$=1, large markers: $k$=5). Results are shown for two source datasets: ImageNet and miniImageNet, each with their corresponding backbones.

with the second largest domain similarity, in the Places dataset with the largest domain similarity, SSL rather exhibits higher CD-FSL accuracy than SL (see the red box in Figure 2). Furthermore, in the ChestX dataset with the smallest domain similarity, SL and SSL have similar performances. These results demonstrate that unlike prior belief, large domain similarity does not always guarantee the superiority of SL. In other words, there is an inconsistency that cannot be explained solely by domain similarity, and we explore why this inconsistency occurs by taking few-shot difficulty into account.

## 5.2 Few-Shot Difficulty

In this sense, we study the impact of few-shot difficulty by categorizing the eight datasets into two groups: one with small domain similarity (*i.e.*, BSCD-FSL) and another with large domain similarity (*i.e.*, other datasets). Figure 3 shows the performance gain of SSL over SL for datasets with varying few-shot difficulty for each group. The performance gain of SSL over SL is defined as $(\mathrm{Error_{sl}} - \mathrm{Error_{ssl}})/\mathrm{Error_{sl}}$, which indicates the relative improvement of the classification error.

OBSERVATION 5.2. *Performance gain of SSL over SL becomes greater at smaller domain similarity or lower few-shot difficulty.*

EVIDENCE. For both groups, the performance gain of SSL over SL becomes greater as few-shot difficulty decreases. In particular, the performance gain is the greatest on the CropDisease and Places datasets with the lowest few-shot difficulty in each group, while the performance gain is the least on the ChestX and CUB datasets with the highest few-shot difficulty in each group. For the target data with higher few-shot difficulty, *it may not be easy to learn discriminative representations by solely using SSL without label supervision*.

Meanwhile, comparing the two groups (BSCD-FSL vs. other datasets), it is observed that the performance gain of SSL over SL is significantly worse for the group with large domain similarity. Namely, the performance gain is near or less than zero when domain similarity is large because features learned from SL with label supervision can be better transferred. Note that the negative value of performance gain means that SL outperforms SSL. Furthermore, the performance gain is closely related to the source dataset size for the datasets with large similarity (see Figures 3(b) and 3(d)). For instance, on the CUB dataset, the performance gain ($k$=5) is $-2.276$ and $-0.249$ for ImageNet and miniImageNet, respectively. However, when domain similarity is small (see Figures 3(a) and 3(c)), the source dataset size does not significantly affect the performance gain of SSL over SL.

In summary, we first conclude that SSL is advantageous to SL when the target domain is extremely dissimilar to the source domain (*i.e.*, the performance gain is greater than 0), which is in line with

Table 3: 5-way 5-shot CD-FSL performance (%) of the models pre-trained by SL, SSL, and MSL including their two-stage versions. ResNet18 is used as the backbone model, and ImageNet is used as the source data for SL and MSL. The balancing coefficient $\gamma$ in Eq. (3) of MSL is set to be 0.875. Datasets are grouped by domain similarity and sorted by few-shot difficulty in ascending order in each group (CropDisease < ChestX | Places < CUB). The best results are marked in bold.

| | Pre-train Scheme | Method | Small Similarity | | | | Large Similarity | | | |
|---|---|---|---|---|---|---|---|---|---|---|
| | | | CropDisease | EuroSAT | ISIC | ChestX | Places | Plantae | Cars | CUB |
| Single-Stage | SL | Default | 92.81±.45 | 84.73±.51 | 44.10±.58 | 25.51±.44 | 79.22±.64 | 63.21±.82 | **66.38**±.80 | **83.93**±.66 |
| | SSL | SimCLR | **97.46**±.34 | **94.12**±.32 | 47.85±.65 | 25.26±.44 | 80.43±.61 | 60.07±.84 | 44.55±.74 | 47.36±.79 |
| | | BYOL | 96.93±.30 | 87.83±.48 | 47.59±.63 | 28.36±.46 | 72.47±.63 | 61.02±.82 | 48.56±.76 | 51.31±.78 |
| | MSL | SimCLR | 96.50±.35 | 90.11±.40 | 45.38±.63 | 26.05±.44 | **82.56**±.58 | 64.76±.83 | 51.84±.79 | 64.53±.80 |
| | | BYOL | 96.74±.31 | 90.82±.40 | **49.14**±.70 | **29.58**±.47 | 81.27±.59 | **67.39**±.81 | 46.76±.73 | 69.67±.82 |

(a) Performance comparison for single-stage schemes.

| | Pre-train Scheme | Method | Small Similarity | | | | Large Similarity | | | |
|---|---|---|---|---|---|---|---|---|---|---|
| Two-Stage | SL→SSL | SimCLR | **97.88**±.30 | **95.28**±.27 | 48.38±.60 | 25.25±.44 | 84.40±.53 | 66.35±.82 | 51.31±.84 | 57.11±.88 |
| | | BYOL | 97.58±.26 | 91.82±.39 | 49.32±.63 | 28.27±.48 | 78.87±.60 | 67.83±.82 | 54.70±.84 | 60.60±.82 |
| | SL→MSL | SimCLR | 97.49±.30 | 91.70±.35 | 47.43±.62 | 26.24±.44 | **85.76**±.52 | 69.24±.81 | 58.97±.82 | 81.51±.72 |
| | | BYOL | 97.09±.31 | 90.89±.40 | **50.72**±.67 | **30.20**±.48 | 83.29±.55 | **74.16**±.77 | 68.87±.80 | 84.34±.67 |
| | SL→MSL⁺ | STARTUP | 96.06±.33 | 89.70±.41 | 46.02±.59 | 27.24±.46 | 85.00±.52 | 69.40±.84 | 68.43±.82 | **89.60**±.55 |
| | | DynDistill | 97.60±.35 | 92.28±.46 | 50.06±.86 | 29.65±.67 | 82.22±.81 | 71.49±1.06 | **69.45**±1.12 | 86.54±1.88 |

(b) Performance comparison for two-stage schemes.

Observation 4.1. This implies supervision with a huge amount of source data cannot overcome domain differences. However, when domain similarity is large, the few-shot difficulty must be considered to determine a better strategy between SSL and SL. Namely, SL becomes more preferable as few-shot difficulty increases due to the benefits from supervision on the source dataset. The same trend is observed when tieredImageNet is used as the source dataset (Appendix I).

## 6 Advanced Scheme: MSL and Two-Stage

In this section, we further study SL and SSL in a more advanced scheme from the domain similarity and few-shot difficulty perspective, in line with previous observations. We first investigate whether SL and SSL can synergize by studying MSL. Next, we analyze the two-stage pre-training scheme used in recent works [47, 30].

### 6.1 Can SL and SSL Synergize?

To identify whether SL and SSL can complement each other, we first consider a mixed-loss pre-training scheme, MSL, described in Eq. (3). We define that synergy between SL and SSL occurs when MSL is superior to both SL and SSL. Table 3(a) summarizes the performance of the models under each pre-training scheme on eight target datasets, grouped by their domain similarity (BSCD-FSL vs. other datasets) and then sorted by the few-shot difficulty in ascending order. In MSL, the hyperparameter $\gamma$ is set to be 0.875 found by a grid search, detailed in Appendix J.

OBSERVATION 6.1. *SL and SSL can synergize when SL and SSL have similar performances.*

EVIDENCE. In Table 3(a), it is observed that SL and SSL can synergize (*i.e.*, MSL > SL, SSL) on four datasets: ISIC, ChestX, Places, and Plantae. SL and SSL have similar performances on these datasets, as shown by the large markers ($k$=5) in Figures 3(a) and 3(b). MSL can learn diverse features, owing to differences in training domains (i.e, source vs. target) and learning frameworks (i.e., supervised vs. unsupervised), which allows for synergy [16, 37, 22, 21]. However, when either SL or SSL significantly outperforms the other, MSL does not perform best. In addition, MSL performance can be improved further in the large similarity group by emphasizing the SL component through a larger batch size (Appendix K).

### 6.2 Extension to Two-Stage Approach

We extend the single-stage to two-stage approaches, extracting more sophisticated target representations. In two-stage pre-training, a model is pre-trained *in prior* with labeled source data in the first

phase and further trained through SSL or MSL in the second phase, *i.e.*, SL → SSL or SL → MSL. This pipeline has been adopted by recent algorithms, such as STARTUP [47] and DynDistill [30], but they additionally maintain an extra network or incorporate the knowledge distillation in the second phase, *i.e.*, SL → MSL$^+$. Table 3(b) summarizes the CD-FSL performance of two-stage schemes.

OBSERVATION 6.2. *Two-stage pre-training schemes are better than their single-stage counterparts.*

EVIDENCE. Two-stage pre-training approaches generally achieve much higher performance than their single-stage counterparts, *i.e.*, SL → SSL outperforms SSL, and SL → MSL outperforms MSL. When SL is used separately in the first phase, it appears to provide a good initialization for the second phase because a converged extractor on the source data is better than a random extractor [40]. Also, the benefit of the two-stage pre-training is significant when domain similarity is large. This observation is promising for practitioners because pre-trained models on ImageNet or bigger datasets are readily accessible. In addition, our simple two-stage methods, without any additional techniques, are shown to achieve comparable performance to the meticulously designed two-stage approaches such as STARTUP, even though our main goal is analysis of basic pre-training methods. Appendix L summarizes the full results including meta-learning based algorithms.

## 7 Conclusion

We established a thorough empirical understanding of CD-FSL. Our work is a pioneering study that unveils hidden findings in the empirical use of CD-FSL. We believe it can inspire subsequent studies like theoretical analysis, which our paper did not cover. In particular, we focused on the effectiveness of SL, SSL, and MSL, which can be realized with single- and two-stage pre-training schemes. We (1) observed that their performances are closely related to domain similarity between the source and target datasets and few-shot difficulty of the target dataset, and (2) proposed how they can be effectively combined for pre-training. Through our empirical study, we presented six findings that have been either misunderstood or unexplored. To justify all the findings, extensive experiments were conducted on benchmarks with varying degrees of domain similarity and few-shot difficulty.

## Acknowledgements

This work was supported by Institute of Information & communications Technology Planning & Evaluation (IITP) grant funded by the Korea government(MSIT) (No.2019-0-00075, Artificial Intelligence Graduate School Program(KAIST), 10%) and Institute of Information & communications Technology Planning & Evaluation(IITP) grant funded by the Korea government(MSIT) (No. 2022-0-00871, Development of AI Autonomy and Knowledge Enhancement for AI Agent Collaboration, 90%)

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
