# A    Self-Supervised Learning Methods

## A.1    SimCLR

SimCLR [3] is one of the simplest yet high-performance contrastive learning methods. Its key idea is mapping the semantically similar examples to be close in the representation space while dissimilar examples to be distant. The similar examples are often called positive samples, and the dissimilar ones are called negative samples. Formally, all examples in the current batch $\{x_k\}_{k=1:B}$ with size $B$ are augmented to generated an augmented batch $\{\tilde{x}_{2k-1}, \tilde{x}_{2k}\}_{k=1:B}$, where $\tilde{x}_{2k-1}$ and $\tilde{x}_{2k}$ are the examples differently augmented from the same input $x_k$. Then, the representations $\{z_{2k-1}, z_{2k}\}_{k=1:B}$ are extracted from a feature extractor with projection layers. Based on the representations, SimCLR performs contrastive learning such that it minimizes the contrastive loss:

$$\mathcal{L}_{\text{SimCLR}} = \frac{1}{2B} \sum_{k=1}^{B} \Big[ \ell(2k-1, 2k) + \ell(2k, 2k-1) \Big]$$

$$\text{where } \ell(i,j) = -\log \frac{\exp(\mathsf{sim}(z_i, z_j)/\tau)}{\sum_{n=1}^{2B} \mathbf{1}_{[n \neq i]} \exp(\mathsf{sim}(z_i, z_n)/\tau)}$$

where $\mathbf{1}$ is an indicator function, $\tau$ is a temperature hyperparameter, and $\mathsf{sim}(\mathbf{u}, \mathbf{v}) = \mathbf{u}^\top \mathbf{v}/\|\mathbf{u}\|\|\mathbf{v}\|$ measures cosine similarity between two vectors $\mathbf{u}$ and $\mathbf{v}$.

## A.2    MoCo

MoCo (Momentum Contrast [26]) is a variant of SimCLR method, which leverages the memory bank and the momentum update of an encoder. Similar to SimCLR, MoCo also minimizes the contrastive loss with positive and negative samples; the positive sample is the other augmentation (view) from the same instance, but the negative samples are not those from the current batch. Instead, MoCo fetches the negative samples from the memory bank, which has been enqueued from the previous batches. To emphasize the usage of the memory bank, the anchor sample, which is contrasted by positive and negative samples, is called query $q$, while the others are called keys $\{k_0, k_1, \ldots k_K\}$. A positive key $k_0$ is the augmentation from the same sample as $q$. Then, MoCo minimizes the following loss:

$$\mathcal{L}_{\text{MoCo}} = -\frac{1}{B} \sum_{i=1}^{B} \log \frac{\exp(\mathsf{sim}(q(i), k_0(i))/\tau)}{\sum_{j=0}^{K} \exp(\mathsf{sim}(q(i), k_j(i))/\tau)}$$

where the query representation and the key representations are extracted from different models. That is, $q = f_q(x_q)$ where $f_q$ is a main encoder, while $k = f_k(x_k)$ where $f_k$ is a momentum encoder that is updated by the moving average of its previous state and that of $f_q$. MoCo overcomes the dependency of the negative sample size on batch size, efficiently achieving the objective of SimCLR using the small memory bank and the additional network.

There are two versions of MoCo: MoCo-v1 [26] and MoCo-v2 [6]. Since MoCo-v2 is a simple improvement of MoCo-v1, such as cosine annealing, MLP projector, and different hyperparameters, we only considered the MoCo-v1 version in this paper.

## A.3    BYOL

While SimCLR and MoCo used positive and negative samples to construct a contrastive task, BYOL (Bootstrap Your Own Latent [23]) achieves higher performances than state-of-the-art contrastive learning models without using the negative samples. That is, BYOL is a non-contrastive SSL method, completely free from the need for negative samples. To this end, BYOL minimizes a similarity loss between the two augmented views using two networks.

There are two networks involved: online network $f_\theta$ and target network $f_\xi$. This is a similar setting to MoCo; the online network is a main encoder and the target network is an encoder that is updated by weighted moving average. Given an image $x$, it augments $x$ into two views $\tilde{x}$ and $\tilde{x}'$. Each view is represented by the encoder with a projector $g_\theta$ and $g_\xi$: $z_\theta = g_\theta(f_\theta(\tilde{x}))$ and $z'_\xi = g_\xi(f_\xi(\tilde{x}'))$. Then, by prediction layers $q_\theta$, a prediction $q_\theta(z_\theta)$ is output and it is compared with the target projection. BYOL uses a mean squared error between the normalized prediction and target projection:

$$\mathcal{L}_{\text{BYOL}} = 2 - 2 \cdot \frac{\langle q_\theta(z_\theta), z'_\xi \rangle}{\|q_\theta(z_\theta)\|_2 \cdot \|z'_\xi\|_2}$$

BYOL also uses a symmetric loss function that passes $\tilde{x}'$ through the online network and $\tilde{x}$ through the target network. The two losses are summed, and the same thing is done for every sample in a batch.

### A.4 SimSiam

SimSiam (Simple Siamese [5]) basically shares a similar idea to the BYOL model. The loss form is exactly the same, but SimSiam does not use an extra target network that is updated by momentum. Instead, SimSiam uses the same online network $f_\theta$ to output the representation of the two views $\tilde{x}, \tilde{x}'$, but blocks the gradient flow for the target projection. While Grill et al. [23] insisted in BYOL on the importance of a momentum encoder since it can prevent collapsing, Chen and He [5] found that a stop-gradient operation is a key to avoiding collapsing. Thus, SimSiam loss is described as follows:

$$\mathcal{L}_{\text{SimSiam}} = 2 - 2 \cdot \frac{\langle q_\theta(z_\theta), \mathsf{sg}(q_\theta(z'_\theta)) \rangle}{\|q_\theta(z_\theta)\|_2 \cdot \|\mathsf{sg}(q_\theta(z'_\theta))\|_2}$$

where $\mathsf{sg}$ indicates the stop-gradient operation.

## B  Datasets Details

### B.1  Datasets

In this paper, we used two source domain datasets and eight target domain datasets. Table 4 summarizes the referenced papers, number of classes, and number of samples of each dataset. For source domain datasets, we used miniImageNet and tieredImageNet, which are two different subsets of the ImageNet-1k dataset [11]. The source dataset for miniImageNet (miniImageNet-train) includes 64 base classes, while the target dataset for miniImageNet (miniImageNet-test) include 20 classes that are disjoint from miniImageNet-train, following Appendix O. Similarly, tieredImageNet is partitioned into a train and test set for the source data and target data, respectively. In our FSL experiments, we also reported the performance of SL model pre-trained on ImageNet. However, we did not actually pre-train with the ImageNet dataset, but fine-tuned from the pre-trained model offered by an official PyTorch [45] library.

The target domain datasets can be separated into two groups: BSCD-FSL benchmark [24] and non-BSCD-FSL. First, the BSCD-FSL benchmark includes CropDisease, EuroSAT, ISIC, and ChestX. These datasets are *supposed* to be distant from the miniImageNet source, with CropDisease most similar and ChestX most dissimilar. The criteria are perspective distortion, semantic content, and color depth. We followed Phoo and Hariharan [47] for the splitting procedure of the target dataset into a pre-training unlabeled set and a few-shot evaluation set. A short description of each dataset is provided below.

- **CropDisease** is a set of diseased plant images.
- **EuroSAT** is a set of satellite images of the landscapes.
- **ISIC** is a set of dermoscopy images of human skin lesions.
- **ChestX** is a set of X-Ray images on the human chest.

In addition to the BSCD-FSL benchmark, we introduced four target datasets that are more commonly used in the (CD-)FSL literature. They are Places, Plantae, Cars, and CUB. However, there is no standard rule to separate the pre-training set and the evaluation set for these four datasets. Thus, we sampled the images from each dataset. A short description and the sampling strategy of a dataset are provided below. Also, for the reproducibility of our work, we provide the code for the sampling procedure and the list of images we used.

- **Places** contains the images designed for scene recognition, such as bedrooms and streets, etc. However, because Places is an enormous dataset to use in the FSL context, we sampled 16 classes out of 365 classes (in a total of train, val, and test). Also, to make the dataset size smaller, we sampled 1,715 images per class, which is a reduced amount from the original 4,941 images per class on average.
- **Plantae** contains the plant images. Similar to Places, we sampled some images to reduce the dataset size. However, unlike Places, Plantae is a highly class-imbalanced set. Therefore, we sampled the top 69 classes that have many samples out of 2,917 classes.

- **Cars** contains the images of 196 car models. We used the entire images that the Cars dataset has (train and test).
- **CUB** contains the images of 200 species of birds. We used the entire images that the CUB dataset has (train, val, and test).

Table 4: Summary of datasets we used in this paper. Note that we used a subset of images for Places and Plantae dataset.

| Datasets | miniImageNet-train | miniImageNet-test | tieredImageNet-train | tieredImageNet-test |
|---|---|---|---|---|
| Reference | Vinyals et al. [66] | Vinyals et al. [66] | Ren et al. [51] | Ren et al. [51] |
| # of classes | 64 | 20 | 351 | 160 |
| # of samples | 38,400 | 12,000 | 448,695 | 206,209 |
| Datasets | CropDisease | EuroSAT | ISIC | ChestX |
| Reference | Mohanty et al. [39] | Helber et al. [27] | Codella et al. [7] | Wang et al. [68] |
| # of classes | 38 | 10 | 7 | 7 |
| # of samples | 43,456 | 27,000 | 10,015 | 25,848 |
| Datasets | Places | Plantae | Cars | CUB |
| Reference | Zhou et al. [74] | Van Horn et al. [65] | Krause et al. [34] | Welinder et al. [70] |
| # of classes | 16 | 69 | 196 | 200 |
| # of samples | 27,440 | 26,650 | 16,185 | 11,788 |

Figure 4 shows the class distribution of each target dataset considered in our study. We observe major differences in the class distributions. For example, the EuroSAT, Places, and CUB datasets have overall balanced class distributions, while the ISIC dataset is extremely unbalanced, with the number of samples per class ranging from 115 to 9,547. We also see that the number of samples per class varies over the eight datasets. The average number of samples per class for the ChestX dataset is 3,693, while for the CUB dataset, this number goes down to only 59.

We posit that the class distribution contributes to the difficulty of each dataset, thus implicitly considered as part of our analysis of target datasets. However, we note that class imbalance is not the deciding factor in dataset difficulty. For example, the CropDisease dataset has a relatively imbalanced class distribution yet is shown to have very low difficulty in our study. Explicitly, the effects of class distribution on CD-FSL have not been studied in our paper.

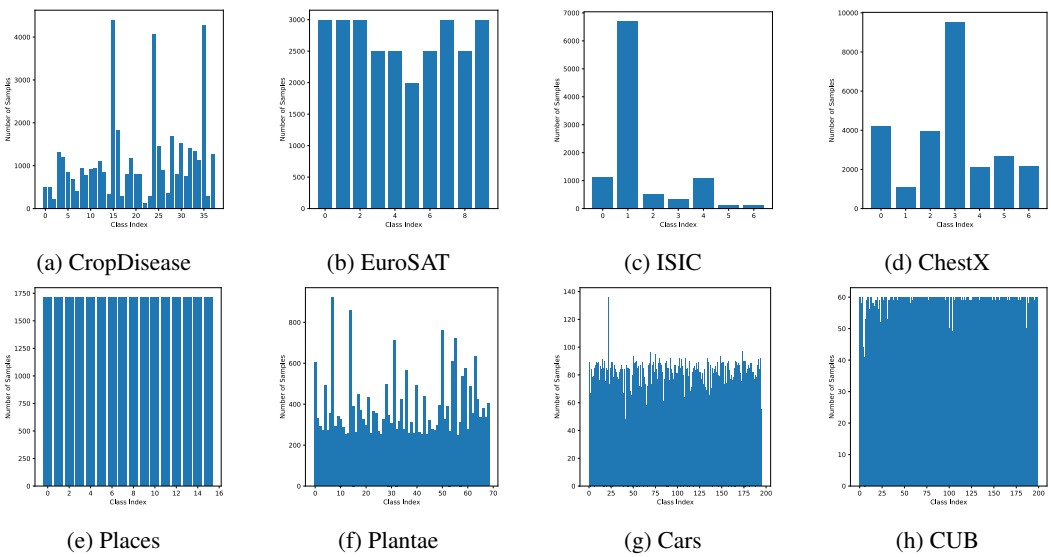

(a) CropDisease     (b) EuroSAT     (c) ISIC     (d) ChestX

(e) Places     (f) Plantae     (g) Cars     (h) CUB

Figure 4: Class distributions of eight target datasets considered in our study.

## B.2  Image Examples

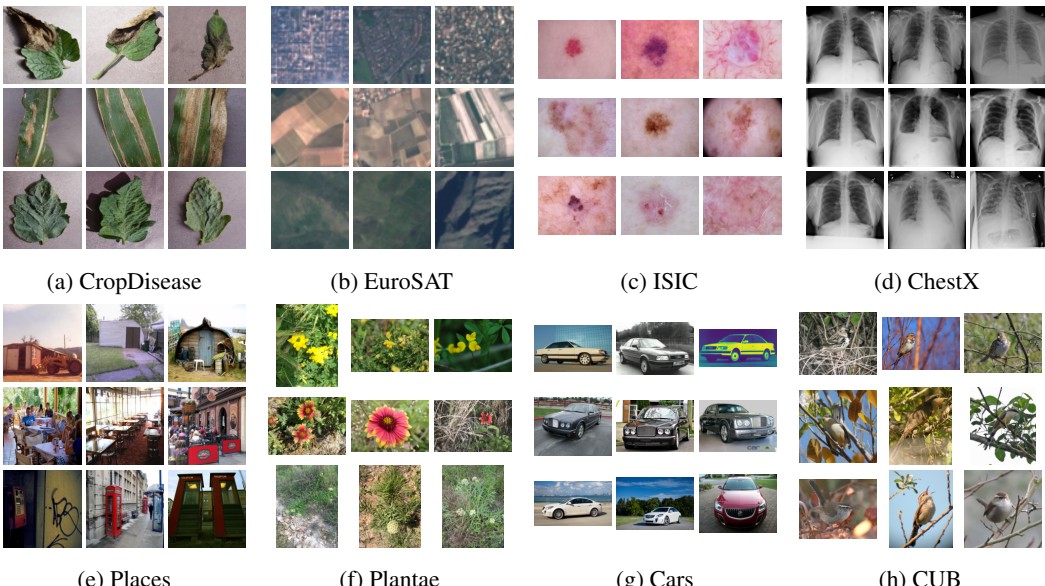

(a) CropDisease  (b) EuroSAT  (c) ISIC  (d) ChestX

(e) Places  (f) Plantae  (g) Cars  (h) CUB

Figure 5: Image examples from eight target datasets considered in our study. Each row displays three samples from a distinct class randomly sampled from each target dataset.

We illustrate the qualitative characteristics of eight target domains for CD-FSL by showing nine randomly sampled examples from three distinct classes for each target dataset in Figure 5. The previous work [24] defined domain similarity to miniImageNet source with respect to perspective distortion, semantic content, and color depth. This can be seen in Figure 5(a)-(d); CropDisease consists of natural images regarding agriculture, EuroSAT contains satellite images taken from a fixed perspective, ISIC and ChestX contain images with fixed perspective and unique semantics, with ChestX being grayscale. On the other hand, non-BSCD-FSL datasets in Figure 5(e)-(h) depict familiar scenes or objects to human eyes.

Using the EMD (Earth Mover's Distance) analysis, we discovered that domain similarity is mainly determined by the semantic content and color depth of the images. For example, we find that ChestX and ISIC, which exhibit highly distinct semantic content, have high domain similarities with all three target datasets. CropDisease and EuroSAT have relatively higher domain similarity within the BSCD-FSL benchmark, and this can be attributed to the fact that the image subjects are from the natural image setting, albeit with fixed perspective and lack of either background or foreground. Places shows the highest domain similarity, which can be attributed to the existence of diverse subjects from the natural image domain, similar to the source datasets.

We can also observe that each target dataset has varying levels of difficulty. For example, for ChestX, it appears challenging to detect the small differences between the grayscale images and nearly impossible to distinguish any prominent features between classes to the untrained eye. On the other hand, classes from CropDisease are shown to have distinct features that are easily distinguishable.

## C  Implementation Details

### C.1  Training Setup

We generally follow the training setup of previous works without validation dataset. Each model was pre-trained on a single RTX A5000, and each pre-training stage of 1000 epochs took 2.5–13.6 hours for ResNet18. SSL pre-training on CropDisease took 6.1, 8.1, 10.3, and 13.6 hours for SimCLR, MoCo, BYOL, and SimSiam, respectively. MSL pre-training took approximately $\times 1.5$ more time compared to SSL, and training time scaled linearly with the size of the target dataset for SSL and MSL.

We explain training details below:

**SL Pre-Training**   We use an SGD optimizer with an initial learning rate of 0.1, the momentum of 0.9, and weight decay coefficient of $10^{-4}$ is used. When using miniImageNet as source data, we train the ResNet10 model for 1,000 epochs with batch size 64, learning rate decayed by 1/10 at epoch {400, 600, 800}. When using tieredImageNet as source data, we train the ResNet18 model for 90 epochs with batch size 256. For ImageNet, the pre-trained ResNet18 model offered by an official PyTorch [45] library is used.

**SimCLR Pre-Training**   We follow the setting in Phoo and Hariharan [47] except batch size, learning rate, and augmentation method; SGD optimizer with momentum 0.9 and weight decay $10^{-4}$ is used. 1000 epochs are trained with batch size 32. Because SimCLR (including other SSL methods) uses a multi-viewed batch, it has an effective batch size of 64 by augmentations. The learning rate starts with 0.1 and is decayed by 1/10 times at epoch {400, 600, 800}. For the SimCLR loss, a two-layer projection head (i.e., Linear-ReLU-Linear) is added on top of the extractor. The projection head uses a hidden dimension of 512 and an output feature dimension of 128. The temperature value of NT-Xent loss (normalized temperature-scaled cross-entropy loss, Chen et al. [3]) is set to 1.0.

**MoCo Pre-Training**   We use the same optimizer, epochs, and batch size as SimCLR pre-training. The projector of both query and key is one fully-connected layer with a feature of dimension 128. Also, we used a moving average coefficient of 0.999 for the momentum encoder and the memory bank size of 1,024. Note that the original MoCo [26] uses a considerable size of a memory bank (i.e., 65,536) because a large number of negative samples is required in self-supervised learning of ImageNet data. However, in the case of our self-supervised learning on small-size target data, a large memory bank is neither needed nor recommended. Moreover, a large number of negative samples can make the contrastive task too hard to optimize for extremely fine-grained images, as we observed in ChestX FSL performances. This is the main reason why MoCo rarely surpasses SimCLR in our experiments. Also, note that the hyperparameters are mainly suited to the SimCLR model, but we did not further search or tune the hyperparameters.

**BYOL Pre-Training**   We use a different optimizer for BYOL; Adam [33] optimizer with the initial learning rate of $3 \times 10^{-4}$. The online and target projector are both composed of two-layer MLP (i.e., Linear-BatchNorm1D-ReLU-Linear) with a hidden dimension of 4,096 and a projection dimension of 256. The moving average coefficient for the target network is 0.99. A predictor after the online projector is also a two-layer MLP with a hidden dimension of 4,096 and an output prediction dimension of 256.

**SimSiam Pre-Training**   SimSiam uses the same network structure as BYOL, except there is no auxiliary target encoder and target projector. Every other training setup is the same.

**MSL Pre-Training**   When training the MSL model, the batch size of source data is 64 and that of target data is 32, because target data are augmented twice to make positive pairs. Although we pre-trained for 1,000 epochs, one epoch corresponds to an entire sweep over the target data. The source batch is randomly sampled at every iteration, independently from the epoch. A conflicting setting is that we used an SGD optimizer for SL pre-training and an Adam optimizer for BYOL pre-training. Therefore, in MSL (BYOL) pre-training, there were two choices of an optimizer. We confirmed with some experiments that the SGD optimizer better works for MSL (BYOL).

**Two-Stage Pre-Training**   In Section 6, we extended the single-stage pre-training to the two-stage approaches. The initial model for the second stage is the SL model, which is exactly the same model as the above **SL Pre-Training**. The second stage of pre-training also follows the same procedure as the single-stage, both for SSL and MSL.

**Fine-Tuning**   We follow the setting in Guo et al. [24]; SGD optimizer with learning rate 0.01, momentum 0.9, and weight decay 0.001 is used. Only the linear classifier is trained with a frozen pre-trained extractor, and 100 epochs are trained with batch size 4. Note that, for a fair comparison, we removed a projector or predictor that is additionally introduced in SSL pre-training.

## C.2 Data Augmentations

We provide below the PyTorch-style code for the base and strong augmentation. A short description for each transform with our set parameter is as follows:

- RandomResizedCrop: Randomly crop a portion of an image and then resize it to 224x224.
- RandomColorJitter: Randomly change the brightness, contrast, and saturation, with a probability of 1.0.
- RandomHorizontalFlip: Randomly flip an image on a vertical axis, with a probability of 0.5.
- RandomGrayscale: Randomly convert image into grayscale, with a probability of 0.1.
- RandomGaussianBlur: Randomly blur an image with Gaussian blur of kernel size (5,5), with a probability of 0.3.

```python
import torchvision.transforms as transforms

def parse_transform(transform, image_size=224):
    if transform == 'RandomResizedCrop':
        return transforms.RandomResizedCrop(image_size)

    elif transform == 'RandomColorJitter':
        return transforms.RandomApply(
                [transforms.ColorJitter(0.4,0.4,0.4,0.0)], p=1.0)

    elif transform == 'RandomGrayscale':
        return transforms.RandomGrayscale(p=0.1)

    elif transform == 'RandomGaussianBlur':
        return transforms.RandomApply(
                [transforms.GaussianBlur(kernel_size=(5,5))], p=0.3)

    elif transform == 'Resize':
        return transforms.Resize([image_size, image_size])

    elif transform == 'Normalize':
        return transforms.Normalize(mean=[0.485,0.456,0.406],
                                    std=[0.229,0.224,0.225])
    elif transform == 'ToTensor':
        return transforms.ToTensor()

def get_composed_transform(augmentation: str, image_size=224):
    if augmentation == 'base':
        transform_list = ['RandomResizedCrop', 'RandomColorJitter',
                          'RandomHorizontalFlip', 'ToTensor', 'Normalize']

    elif augmentation == 'strong':
        transform_list = ['RandomResizedCrop', 'RandomColorJitter',
                          'RandomGrayscale', 'RandomGaussianBlur',
                          'RandomHorizontalFlip', 'ToTensor', 'Normalize']

    elif augmentation == 'none':
        transform_list = ['Resize', 'ToTensor', 'Normalize']

    transform_funcs = [parse_transform(x, image_size=image_size)
                        for x in transform_list]
    transform = transforms.Compose(transform_funcs)
    return transform
```

## D  Domain Similarity

To estimate the domain similarity, we follow Cui et al. [10] and Li et al. [36] by calculating EMD as the distance between two domains. EMD is informally defined as the minimum cost of moving one accumulation into another. EMD has advantages compared to other metric choices, such as Kullback-Leibler divergence (KLD), Jensen-Shannon divergence (JSD), or maximum mean discrepancy (MMD). We can compute EMD directly from the samples, whereas KLD and JSD require explicit expressions for the densities [9]. MMD can also be considered but is less powerful in high dimensions and highly dependent on the kernel and its hyperparameters [49].

The BSCD-FSL benchmark contains four datasets with varying levels of domain similarity: CropDisease, EuroSAT, ISIC, and ChestX. These datasets are known to be distant from the source dataset, miniImageNet. Guo et al. [24] provided the order of domain similarity for the BSCD-FSL benchmark based on three qualitative factors: perspective distortion, semantic contents, and color depth. However, our quantitative metric in Eq. (4) shows a somewhat different order of domain similarity between the four datasets in BSCD-FSL. The known similarity order for BSCD-FSL was *"CropDisease > EuroSAT > ISIC > ChestX"* under the assumption that a dataset has domain similar to ImageNet if it has perspective distortion (*i.e.*, CropDisease), is natural (*i.e.*, CropDisease, EuroSAT), and has RGB color depth (*i.e.*, CropDisease, EuroSAT, ISIC).

In contrast, we observe a different order of *"EuroSAT > CropDisease ≈ ISIC > ChestX"* in Table 5 and Table 6 when using our quantitative metric. It turns out that semantic content and color depth are significant factors in deciding domain similarity; thus, ChestX is always the most dissimilar to the source domain. On the other hand, perspective distortion is less important than semantic content and color depth for determining domain similarity, considering that the order of CropDisease and EuroSAT is reversed. Therefore, we observe that EuroSAT is the closest dataset to the source domain. The change in the domain similarity rank of EuroSAT, when tieredImageNet is used as the source dataset, is discussed below.

Table 5: Domain Similarity. Earth Mover's Distance (EMD) and similarity (calculated by $\exp(-\alpha \times \text{EMD})$) are reported. The feature extractor used is ResNet101 provided by PyTorch [45]. Rank 1 dataset indicates that the source and target datasets are the most similar.

|  |  | Places | CUB | Cars | Plantae | EuroSAT | CropDisease | ISIC | ChestX |
|---|---|---|---|---|---|---|---|---|---|
| EMD | IN | 18.14 | 18.91 | 20.13 | 20.31 | 21.67 | 21.88 | 22.25 | 22.28 |
|  | tieredIN | 17.26 | 19.90 | 20.23 | 20.63 | 19.20 | 22.07 | 22.94 | 23.19 |
|  | miniIN | 17.49 | 19.38 | 20.34 | 20.29 | 21.10 | 21.66 | 22.20 | 22.33 |
| Sim | IN | 0.834 (1) | 0.828 (2) | 0.818 (3) | 0.816 (4) | 0.805 (5) | 0.803 (6) | 0.801 (7) | 0.800 (8) |
|  | tieredIN | 0.841 (1) | 0.820 (3) | 0.817 (4) | 0.814 (5) | 0.825 (2) | 0.802 (6) | 0.795 (7) | 0.793 (8) |
|  | miniIN | 0.840 (1) | 0.824 (2) | 0.816 (4) | 0.816 (3) | 0.810 (5) | 0.805 (6) | 0.801 (7) | 0.800 (8) |

Domain similarity can differ according to the feature extractor used because it is based on representations. In the main paper, we use ResNet101 to extract representations because we use ResNet-like models for our few-shot classification tasks. However, other architectures (*e.g.*, DenseNet and ViT) can also be used. Table 6 shows the domain similarity to the ImageNet source dataset, measured using different feature extractors. To do this, we used the open-source library `timm`.[6] For the details of each architecture, please refer to the original papers: ResNet [25], MobileNetV2 [53], EfficientNet [59, 60], DenseNet [29], and ViT [15].

Although the exact ordering of domain similarity can change, it does not undermine the consistency of our analysis. In particular, we explain that:

- **Important point for Obs. 5.1.** From Obs. 5.1, we argue that larger domain similarity does not always guarantee the superiority of SL. To demonstrate this, we compare the performance of SL and SSL on Places and CUB. Namely, SSL is better than SL on Places, while SL is better than SSL on CUB, despite Places being closer to the source dataset. As we can see in Table 6, this observation does not change, even when using other feature extractors. In fact, when EfficientNet-b4 is used, the domain similarity ranking of Cars and EuroSAT are changed, exacerbating the inconsistency. Furthermore, when using ViT models, the similarity ranking of CUB moves down to fifth place,

---

[6] http://github.com/rwightman/pytorch-image-models/

despite it showing the largest margin between SL and SSL performance, in favor of SL on source data (refer to Figure 2).

- **Important point for Obs. 5.2.** From Obs. 5.2, we argue that for both groups, the performance gain of SSL over SL becomes greater as few-shot difficulty decreases. We first divide the eight target datasets into two groups according to the domain similarity. Within each similarity group, the performance gain of SSL over SL is highly related to the few-shot difficulty. As shown in Table 6, EuroSAT can be categorized into the large similarity group when using EfficientNet-b4, ViT-B/16, or ViT-L/16. However, because EuroSAT has lower few-shot difficulty than Places (refer to Figure 1), the superiority of SSL over SL on EuroSAT is consistently explained with few-shot difficulty, even when inside the large similarity group. In addition, the domain similarity ranking of CropDisease and ISIC based on ResNet101 is different from that based on other extractors. However, both datasets remain inside the small similarity group, hence does not affect Obs. 5.2.

Table 6: Domain Similarity to ImageNet measured across different architectures. Similarities (calculated by $\exp(-\alpha \times \text{EMD})$) are reported. The feature extractors used are ResNet101, provided by PyTorch [45], and others, by `timm` open-source library. Rank 1 dataset indicates that the source and target datasets are the most similar.

| Extractor | Places | CUB | Cars | Plantae | EuroSAT | CropDisease | ISIC | ChestX |
|---|---|---|---|---|---|---|---|---|
| ResNet101 (main) | 0.834 (1) | 0.828 (2) | 0.818 (3) | 0.816 (4) | 0.805 (5) | 0.803 (6) | 0.801 (7) | 0.800 (8) |
| ResNet18 | 0.866 (1) | 0.847 (2) | 0.845 (3) | 0.843 (4) | 0.829 (5) | 0.823 (7) | 0.828 (6) | 0.815 (8) |
| MobileNetV2 | 0.919 (1) | 0.918 (2) | 0.908 (4) | 0.910 (3) | 0.903 (5) | 0.889 (7) | 0.893 (6) | 0.884 (8) |
| EfficientNet-b0 | 0.913 (1) | 0.910 (2) | 0.903 (3) | 0.901 (4) | 0.901 (5) | 0.873 (7) | 0.877 (6) | 0.873 (8) |
| EfficientNet-b4 | 0.969 (1) | 0.967 (2) | 0.963 (5) | 0.965 (4) | 0.966 (3) | 0.956 (7) | 0.961 (6) | 0.953 (8) |
| EfficientNetV2 | 0.930 (1) | 0.927 (2) | 0.924 (3) | 0.924 (4) | 0.924 (5) | 0.906 (7) | 0.914 (6) | 0.896 (8) |
| DenseNet121 | 0.818 (1) | 0.802 (2) | 0.791 (3) | 0.787 (4) | 0.785 (5) | 0.761 (7) | 0.767 (6) | 0.752 (8) |
| ViT-B/16 | 0.508 (1) | 0.415 (5) | 0.438 (4) | 0.442 (3) | 0.444 (2) | 0.386 (7) | 0.409 (6) | 0.390 (8) |
| ViT-L/16 | 0.478 (1) | 0.395 (5) | 0.396 (4) | 0.426 (2) | 0.422 (3) | 0.372 (7) | 0.391 (6) | 0.367 (8) |

# E  Few-Shot Difficulty

We quantify few-shot difficulty using our empirical upper bound on each dataset following in Eq. (5). The few-shot difficulty depends on a backbone network and $k$. Table 7 describes few-shot difficulty according to the combination of our backbone (ResNet10 and ResNet18) and $k$. It is observed that the order of few-shot difficulty remains the same, except between ISIC and CUB when ResNet10 is used as a backbone and $k$=1. We point out that Obs. 5.2 still stands under this variation.

Table 7: Few-shot difficulty (ranking). 5-way $k$-shot performances are reported. To quantify the data difficulty, we designed the upper performance case, where we use SL pre-training with 20% of target data as labeled data. The $k$-shot difficulty is calculated by Diff@$k = \exp(-\beta \times \text{Perf@}k)$. Rank 1 dataset is the most difficult one.

| Backbone | $k$ | CropDisease | EuroSAT | ISIC | ChestX | Places | Plantae | Cars | CUB |
|---|---|---|---|---|---|---|---|---|---|
| | 1 | 96.92±.32 | 90.51±.55 | 42.83±.80 | 31.00±.60 | 63.97±.87 | 52.83±.89 | 48.71±.82 | 42.96±.76 |
| RN18 | 5 | 99.51±.10 | 96.74±.21 | 55.55±.67 | 39.19±.58 | 81.56±.57 | 74.24±.71 | 72.83±.67 | 64.03±.77 |
| | 20 | 99.69±.07 | 97.45±.17 | 61.32±.62 | 42.11±.56 | 86.10±.47 | 82.17±.64 | 82.08±.53 | 74.14±.66 |
| Diff@1 | | 0.379 (8) | 0.405 (7) | 0.652 (2) | 0.733 (1) | 0.527 (6) | 0.590 (5) | 0.614 (4) | 0.651 (3) |
| Diff@5 | | 0.370 (8) | 0.380 (7) | 0.574 (2) | 0.676 (1) | 0.442 (6) | 0.476 (5) | 0.483 (4) | 0.527 (3) |
| Diff@20 | | 0.369 (8) | 0.377 (7) | 0.542 (2) | 0.656 (1) | 0.423 (6) | 0.440 (5) | 0.440 (4) | 0.476 (3) |
| | 1 | 92.44±.55 | 83.34±.68 | 42.89±.77 | 28.89±.55 | 57.25±.82 | 49.08±.83 | 43.32±.72 | 40.72±.73 |
| RN10 | 5 | 99.00±.15 | 95.37±.26 | 56.94±.65 | 36.59±.56 | 77.39±.63 | 70.56±.76 | 66.40±.68 | 60.29±.78 |
| | 20 | 99.54±.07 | 97.28±.18 | 63.93±.58 | 42.03±.55 | 84.62±.49 | 80.50±.65 | 78.56±.57 | 71.71±.67 |
| Diff@1 | | 0.397 (8) | 0.435 (7) | 0.651 (3) | 0.749 (1) | 0.564 (6) | 0.612 (5) | 0.648 (4) | 0.666 (2) |
| Diff@5 | | 0.372 (8) | 0.385 (7) | 0.566 (2) | 0.694 (1) | 0.461 (6) | 0.494 (5) | 0.515 (4) | 0.547 (3) |
| Diff@20 | | 0.370 (8) | 0.378 (7) | 0.528 (2) | 0.657 (1) | 0.429 (6) | 0.447 (5) | 0.456 (4) | 0.488 (3) |

**Few-shot Difficulty on Different Splits.**   We used the same 20% split of $\mathcal{D}_U$ as used in SSL pre-training for measuring the few-shot difficulty, but with label information. This is because the dataset partition for calculating few-shot difficulty (which is the rest 80%) should be matched with that for evaluating SL/SSL methods, for consistent analysis. However, few-shot difficulty can differ according to the dataset splits. To remedy this concern, we provide the few-shot performance using different splits when 20% of target dataset are used for pre-training with label information. Table 8 shows that the ranks of few-shot difficulty between datasets do not change even if dataset splits are changed.

Table 8: 5-way $k$-shot performances are reported. These performances are converted to few-shot difficulty. To quantify the data difficulty, we designed the upper performance case, where we use SL pre-training with 20% of target data as labeled data. To show the robustness of few-shot difficulty, accuracy is estimated three times using different splits for 20% of target data. ResNet18 is used as a backbone.

| Split seed | $k$ | CropDisease | EuroSAT | ISIC | ChestX | Places | Plantae | Cars | CUB |
|---|---|---|---|---|---|---|---|---|---|
| | 1 | 96.92±.32 | 90.51±.55 | 42.83±.80 | 31.00±.60 | 63.97±.87 | 52.83±.89 | 48.71±.82 | 42.96±.76 |
| 1 | 5 | 99.51±.10 | 96.74±.21 | 55.55±.67 | 39.19±.58 | 81.56±.57 | 74.24±.71 | 72.83±.67 | 64.03±.77 |
| | 20 | 99.69±.07 | 97.45±.17 | 61.32±.62 | 42.11±.56 | 86.10±.47 | 82.17±.64 | 82.08±.53 | 74.14±.66 |
| | 1 | 96.52±.36 | 90.84±.50 | 43.05±.74 | 30.03±.62 | 63.61±.90 | 54.94±.88 | 48.98±.85 | 43.84±.80 |
| 2 | 5 | 99.51±.09 | 96.93±.20 | 55.92±.65 | 39.64±.55 | 81.65±.56 | 75.73±.72 | 72.95±.63 | 63.76±.78 |
| | 20 | 99.78±.05 | 97.67±.16 | 63.34±.56 | 45.96±.54 | 86.18±.45 | 82.74±.59 | 81.41±.49 | 75.41±.67 |
| | 1 | 96.54±.34 | 89.23±.56 | 42.94±.78 | 29.36±.60 | 65.25±.84 | 54.20±.91 | 48.39±.79 | 43.95±.75 |
| 3 | 5 | 99.54±.08 | 96.73±.19 | 56.78±.68 | 38.23±.57 | 82.44±.54 | 74.51±.79 | 71.71±.70 | 64.39±.75 |
| | 20 | 99.81±.05 | 97.60±.16 | 63.16±.62 | 44.60±.55 | 86.64±.45 | 82.94±.61 | 82.04±.50 | 75.79±.64 |

# F Domain Similarity and Few-Shot Difficulty Visualizations

In this section, we provide visualizations of domain similarity and few-shot difficulty. Domain similarity is dependent on the source dataset, and few-shot difficulty is dependent on backbone network (*e.g.*, ResNet10 and ResNet18) and $k$. Figure 6 visualizes domain similarity and few-shot difficulty for eight datasets, as depicted in Appendix D and E. Figure 6(a,b,c) have the same domain similarity, Figure 6(d,e,f) have the same, and Figure 6(g,h,i) have the same, because domain similarity is based on the source dataset. For few-shot difficulty, Figure 6(a,d) have the same difficulty, 6(b,e) have the same, and (c,f) have the same, because few-shot difficulty is based on backbone network and $k$.

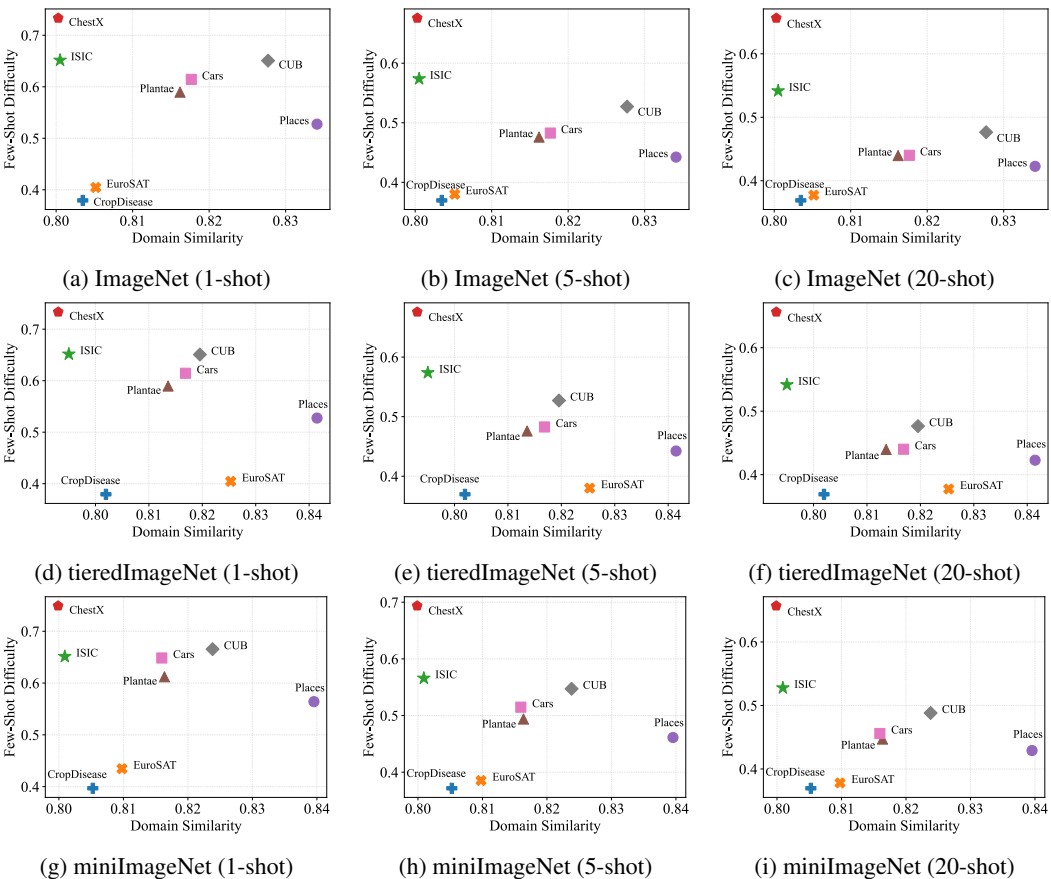

Figure 6: Domain similarity and few-shot difficulty for eight benchmark datasets.

# G    Performance of SSL according to the Ratio of Unlabeled Target Data

In this section, we evaluate few-shot performance of SSL (SimCLR and BYOL) according to the ratio of unlabeled target data when ResNet10 is used as a backbone network. Figure 7 and Figure 8 describe the few-shot performance according to the ratio of unlabeled target data when SimCLR and BYOL are used for SSL method, respectively. We control the ratio $\in$ {5%, 10%, 20%, 40%, 80%}. We further evaluate few-shot performance of SSL (SimCLR) when ResNet18 is used as a backbone network, depicted as Figure 9.

It is observed that except for ChestX, SimCLR with a small portion (even 5%) of target data as unlabeled data has better performance than SL that uses ImageNet, tieredImageNet, and miniImageNet. Note that ImageNet and tieredImageNet include around 1.3 million and 0.45 million samples with annotations, respectively. On the other hand, 5% of EuroSAT, CropDisease, and ISIC unlabeled data include around 1.4k, 2.2k, and 0.5k samples. It implies that the consistency between source and target domains is much more important than the number of data for pre-training.

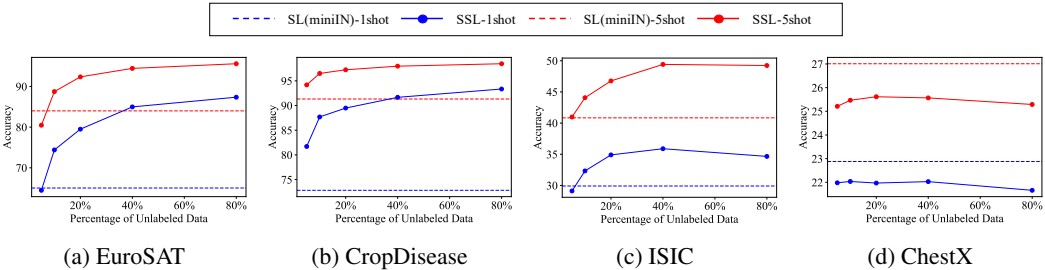

Figure 7: 5way-$k$shot performance of SSL (SimCLR) according to the ratio of unlabeled target data and SL (Section 4). ResNet10 is used as a backbone. Blue and red lines indicate 1-shot and 5-shot accuracy, respectively. Dotted and solid lines are accuracy of SL (miniIN) and SSL (SimCLR), respectively.

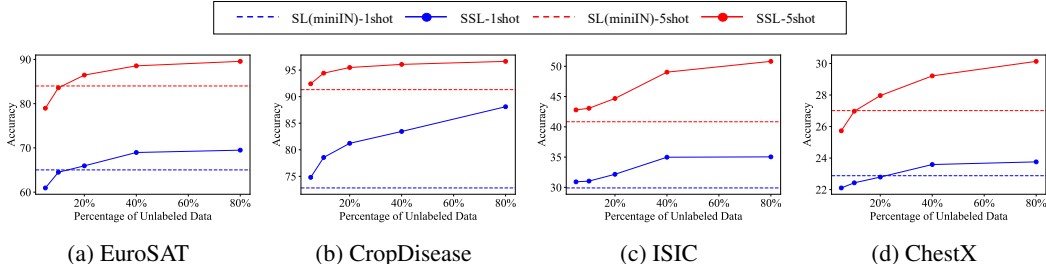

Figure 8: 5way-$k$shot performance of SSL (BYOL) according to the ratio of unlabeled target data and SL (Section 4). ResNet10 is used as a backbone. Blue and red lines indicate 1-shot and 5-shot accuracy, respectively. Dotted and solid lines are accuracy of SL (miniIN) and SSL (BYOL), respectively.

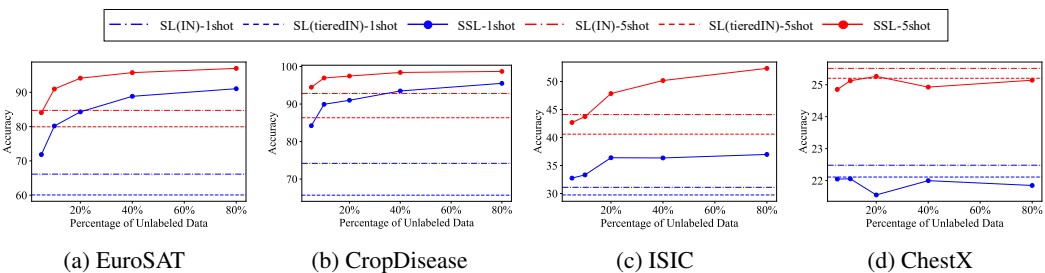

Figure 9: 5way-$k$shot performance of SSL according to the ratio of unlabeled target data and SL (Section 4). ResNet18 is used as a backbone. SimCLR is used for the SSL method.

## H  Performance of SL and SSL for the Other Datasets

Table 9 summarizes the few-shot performance of SL and SSL on non-BSCD-FSL datasets. Note that these datasets are known to be closer to the ImageNet than BSCD-FSL datasets [16] and our estimated similarity shows the same trend. We would like to highlight that unlike BSCD-FSL sets, these four target domains take a big advantage from the ImageNet dataset. SL pre-trained on ImageNet has comparable or even better (in Cars and CUB) performance than SSL.

Table 9: 5-way $k$-shot CD-FSL performance of the models pre-trained by SL and SSL, on four additional target datastes: Places, Plantae, Cars, and CUB. We report the average accuracy and its 95% confidence interval over 600 few-shot episodes. B and S indicate base and strong augmentations, respectively. The best results are marked in bold and the second best are underlined. We include the result when using the model pre-trained on the entire ImageNet data, which also uses the ResNet18 backbone as tieredImageNet experiments.

| Source Data | Pre-train Scheme | Method | Aug. | Places | | Plantae | | Cars | | CUB | |
|---|---|---|---|---|---|---|---|---|---|---|---|
| | | | | $k$=1 | $k$=5 | $k$=1 | $k$=5 | $k$=1 | $k$=5 | $k$=1 | $k$=5 |
| ImageNet | SL | Default | B | 57.47±.86 | 79.22±.64 | 43.66±.80 | **63.21**±.82 | **45.82**±.79 | **66.38**±.80 | **65.24**±.97 | **83.93**±.66 |
| tiered ImageNet | SL | Default | B | 52.07±.86 | 72.12±.69 | 38.63±.74 | 54.76±.82 | 31.23±.65 | 42.59±.70 | 57.94±.93 | 76.86±.78 |
| | | | S | 52.82±.86 | 72.96±.67 | 34.99±.64 | 51.11±.76 | 31.05±.63 | 42.32±.69 | 54.18±.91 | 74.14±.80 |
| Target Data | SSL | SimCLR | B | 45.82±.85 | 62.07±.78 | 38.52±.74 | 53.89±.80 | 28.86±.68 | 37.05±.69 | 33.56±.67 | 43.99±.71 |
| | | | S | **64.97**±.94 | **80.43**±.61 | **44.18**±.85 | 60.07±.84 | 32.46±.70 | 44.55±.74 | 36.15±.76 | 47.36±.79 |
| | | MoCo | B | 39.64±.82 | 53.95±.77 | 35.17±.73 | 48.83±.76 | 27.40±.64 | 34.59±.67 | 29.67±.59 | 36.93±.61 |
| | | | S | 55.53±.74 | 71.50±.73 | 36.49±.73 | 49.15±.76 | 29.36±.67 | 38.44±.70 | 31.76±.66 | 40.81±.72 |
| | | BYOL | B | 40.38±.72 | 60.06±.73 | 38.60±.72 | 57.81±.81 | 31.04±.66 | 41.79±.72 | 35.27±.67 | 49.61±.71 |
| | | | S | 51.76±.79 | 72.47±.63 | 42.16±.75 | 61.02±.82 | 34.54±.70 | 48.56±.76 | 36.50±.68 | 51.31±.78 |
| | | SimSiam | B | 35.27±.68 | 48.12±.69 | 36.11±.76 | 48.63±.79 | 28.30±.64 | 35.24±.65 | 29.96±.62 | 37.61±.60 |
| | | | S | 52.56±.92 | 68.29±.74 | 36.19±.69 | 50.23±.76 | 31.21±.64 | 43.06±.67 | 33.73±.71 | 43.22±.74 |
| (a) ResNet18 is used as a backbone. | | | | | | | | | | | |
| mini ImageNet | SL | Default | B | 51.84±.80 | 72.19±.70 | 37.28±.69 | 54.15±.74 | 30.79±.56 | 44.36±.69 | **40.65**±.78 | **58.54**±.81 |
| | | | S | 52.45±.78 | 72.92±.66 | 36.72±.67 | 53.26±.73 | 30.20±.54 | 44.39±.66 | 40.56±.78 | 58.10±.78 |
| Target Data | SSL | SimCLR | B | 44.06±.78 | 62.86±.78 | 38.43±.77 | 54.68±.80 | 28.59±.66 | 38.24±.73 | 33.88±.68 | 45.31±.73 |
| | | | S | **58.75**±.93 | **78.39**±.61 | **42.65**±.80 | **59.77**±.82 | 30.89±.66 | **45.60**±.72 | 35.49±.73 | 47.69±.77 |
| | | MoCo | B | 38.41±.74 | 54.65±.74 | 33.96±.69 | 47.51±.72 | 28.03±.66 | 36.19±.72 | 32.37±.65 | 40.55±.72 |
| | | | S | 52.05±.90 | 71.57±.70 | 36.36±.73 | 50.37±.78 | 28.25±.61 | 38.89±.69 | 33.53±.72 | 42.87±.74 |
| | | BYOL | B | 40.60±.69 | 59.28±.71 | 39.27±.73 | 55.87±.79 | 30.11±.62 | 41.21±.69 | 34.74±.65 | 49.10±.74 |
| | | | S | 47.81±.75 | 68.14±.68 | 39.12±.71 | 55.31±.79 | **31.53**±.65 | 43.92±.70 | 35.96±.70 | 49.34±.76 |
| | | SimSiam | B | 39.27±.72 | 53.40±.74 | 37.12±.72 | 50.61±.81 | 28.49±.62 | 35.50±.67 | 30.37±.63 | 38.22±.62 |
| | | | S | 51.62±.81 | 69.77±.66 | 38.49±.73 | 53.10±.78 | 30.00±.59 | 40.92±.67 | 34.25±.71 | 44.85±.74 |
| (b) ResNet10 is used as a backbone. | | | | | | | | | | | |

# I Analyses on Other Source Datasets

In this section, we expand our analyses of SL and SSL on ImageNet source in Section 5, onto two additional source datasets: tieredImageNet and miniImageNet. Every observation that was previously identified on ImageNet is consistently made in the two additional source datasets.

## I.1 Limitations of Domain Similarity (Observation 5.1)

Figure 10 shows the performance of SL and SSL for three source datasets and eight target datasets, according to domain similarity. Across all source datasets, we consistently find that domain similarity alone is not sufficient to explain the relative performance of SL, compared to SSL. As mentioned in Section 5, we observe that SSL can outperform SL even when domain similarity is large, as highlighted by the difference between Places and CUB shown in Figures 10(a,b) for ImageNet, which are identical to Figures 2(a,b) in the main paper. Similar observations are made between EuroSAT and CUB for tieredImageNet in Figures 10(c,d), and between Places and CUB for miniImageNet in Figures 10(e,f).

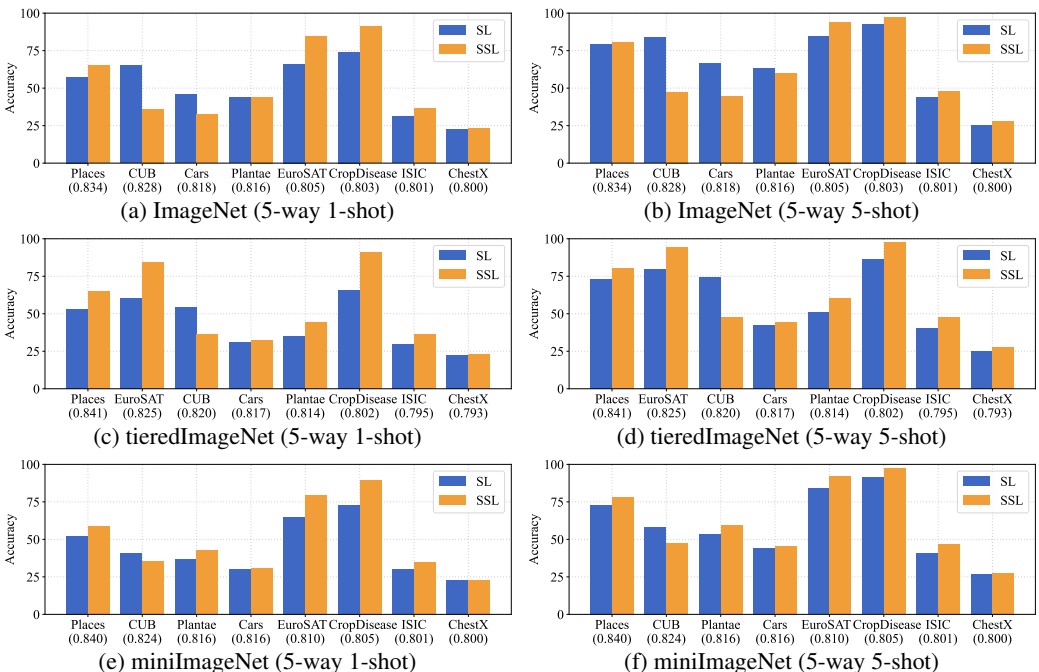

Figure 10: 5-way $k$-shot CD-FSL performance (%) of SL and SSL according to domain similarity. Target datasets are shown in order of domain similarity (values in x-axis) to ImageNet, tieredImageNet and miniImageNet, respectively. For SSL, SimCLR is used for all datasets except ChestX, for which BYOL is used.

## I.2 When Does Performance Gain of SSL over SL Become Greater? (Observation 5.2)

Figure 11 shows the performance gain of SSL over SL for three source datasets and eight target datasets, according to few-shot difficulty, for two groups with different levels of domain similarity. Again, the identical observation is made for all three source datasets. When comparing the two groups (BSCD-FSL vs. others), larger performance gain is observed for the small domain similarity group (BSCD-FSL), compared to the latter (others). Within each group, the performance gain of SSL over SL increases with lower few-shot difficulty.

In addition, comparing between different source datasets, for target datasets with large similarity (Figure 11(b,d,f)), the performance gain of SSL over SL decreases by larger source dataset size. For example, on the CUB dataset, the performance gain (for $k = 5$) is $-0.249$, $-1.035$, and $-2.276$ for miniImageNet, tieredImageNet, and ImageNet, respectively. However, for target datasets with small similarity (Figure 11(a,c,e)), the performance gain of SSL over SL does not have a consistent trend according to the source dataset size.

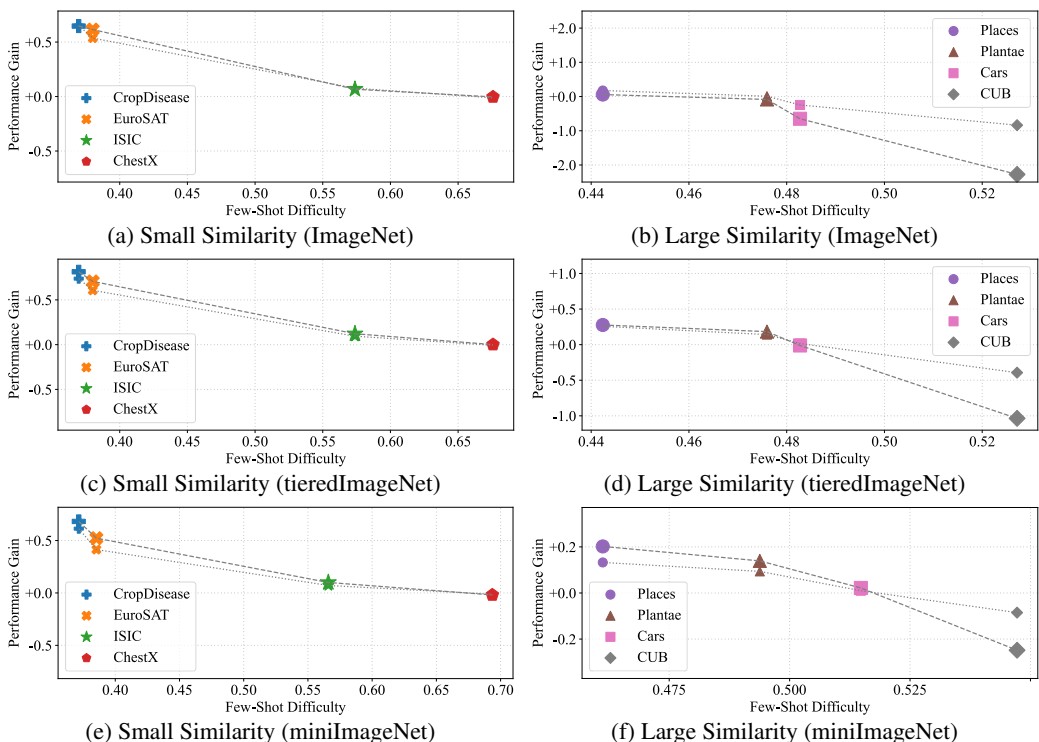

Figure 11: 5-way $k$-shot performance gains of SSL over SL for the two dataset groups according to the few-shot difficulty (small: $k$=1, large: $k$=5). Results are shown for three source datasets: ImageNet, tieredImageNet, and miniImageNet, each with their corresponding backbones. SimCLR is used for SSL in all target datasets except ChestX, for which BYOL is used.

# J  Hyperparameter $\gamma$ in MSL Pre-Training

## J.1  Choice of Hyperparameter $\gamma$

One important hyperparameter in MSL is a balancing weight $\gamma$ (refer to Eq. (3)). We investigated how we should choose $\gamma$ value. Figure 12 and Figure 13 describe the few-shot performance of MSL according to the balancing weight $\gamma$ between SL and SSL when SimCLR or BYOL are used for SSL, respectively. In Figure 12, MSL performance (circle-marked solid lines) generally improves as $\gamma$ increases from 0.125 to 0.875, i.e., the weight for SSL is getting larger, except for ChestX. In Section 4, we found that non-contrastive SSL method nicely worked on ChestX. Figure 13 shows that MSL with BYOL loss guarantees good performance on ChestX in $\gamma = 0.875$. We further increased $\gamma$ to {0.9, 0.95, 0.99}, but there was an overall decreasing trend of accuracy, so we fixed $\gamma$ to 0.875 in every MSL experiment in the paper.

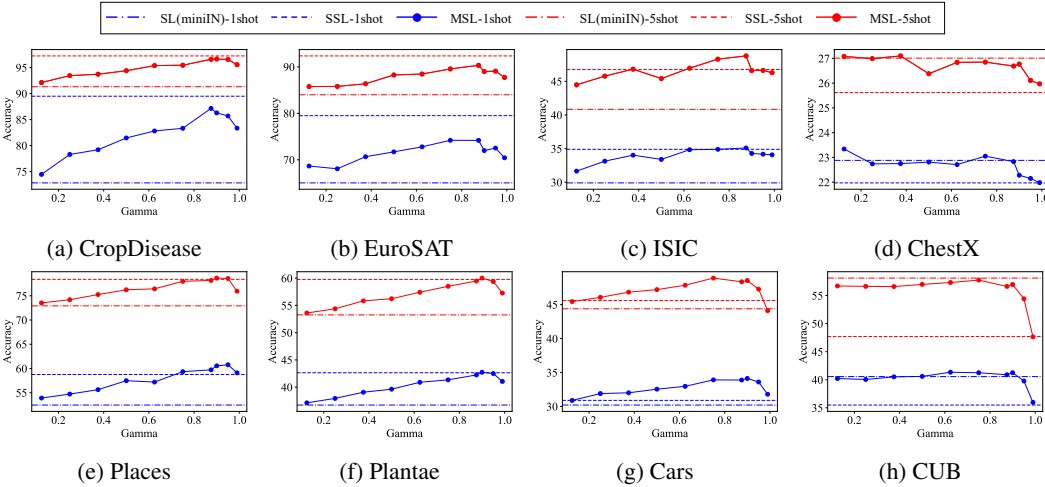

Figure 12: 5-way $k$-shot performance of MSL according to the balancing weight (i.e., $\gamma$) between SL and SSL (Section 6). ResNet10 is used as a backbone. SimCLR is used for the MSL and SSL method.

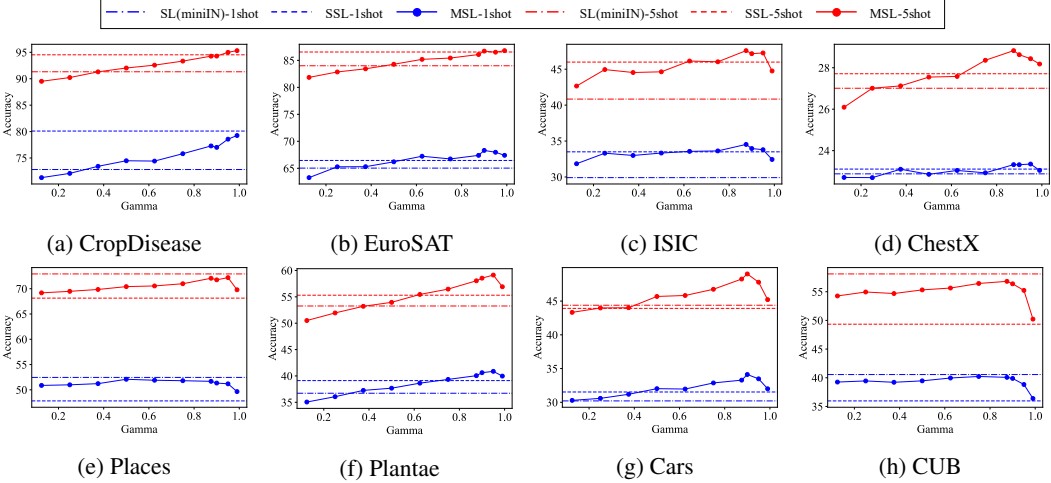

Figure 13: 5-way $k$-shot performance of MSL according to the balancing weight (i.e., $\gamma$) between SL and SSL (Section 6). ResNet10 is used as a backbone. BYOL is used for the MSL and SSL method.

## J.2 Dynamic Hyperparameter $\gamma$

Inspired by the two-stage pre-training schemes in Section 6, we investigate the effects of dynamically increasing $\gamma$ during the single-stage pre-training. Specifically, we investigate a simple pre-training scheme in which $\gamma$ linearly increases from 0 to 1 over the course of 1000 epochs (*i.e.,* single-stage MSL with $\gamma = 0 \nearrow 1$).

Table 10 and Table 11 describe the few-shot performance of the devised method (in the lowermost row), with performances of other methods displayed for ease of comparison. We observe that the performance of the devised method lies between that of standalone SL and SSL except for ChestX. The devised method typically underperforms two-stage pre-training as well as standalone MSL, indicating that it is not an effective method to exploit both SL and SSL dynamically.

Table 10: 5-way 1-shot CD-FSL performance (%) of the models pre-trained with varying configurations of $\gamma$ in Eq. (3) of MSL. ResNet18 is used as the backbone model, and ImageNet is used as the source data for SL. The best results are marked in bold.

| Pre-train Scheme | $\gamma$ | Method | Small Similarity | | | | Large Similarity | | | |
| --- | --- | --- | --- | --- | --- | --- | --- | --- | --- | --- |
| | | | CropDisease | EuroSAT | ISIC | ChestX | Places | Plantae | Cars | CUB |
| SL | 0 | Default | 74.18±.82 | 66.14±.83 | 31.11±.55 | 22.48±.39 | 57.47±.86 | 43.66±.80 | 45.82±.79 | **65.24**±.97 |
| SSL | 1 | SimCLR | 91.00±.76 | 84.30±.73 | 36.39±.66 | 21.55±.41 | 64.97±.94 | 44.18±.85 | 32.46±.70 | 36.15±.76 |
| | | BYOL | 85.77±.73 | 66.16±.86 | 34.53±.62 | 22.75±.41 | 51.76±.79 | 42.16±.75 | 34.54±.70 | 36.50±.68 |
| MSL | 0.875 | SimCLR | 88.38±.70 | 73.97±.79 | 34.02±.62 | 22.04±.40 | 65.13±.88 | 47.47±.86 | 36.96±.77 | 47.35±.87 |
| | | BYOL | 86.47±.74 | 73.18±.83 | **37.10**±.67 | 23.97±.44 | 61.40±.87 | 48.31±.86 | 33.31±.66 | 50.71±.87 |
| SL → SSL | 0→1 | SimCLR | **92.24**±.70 | **86.51**±.67 | 36.11±.67 | 21.75±.41 | **71.05**±.92 | 49.02±.91 | 37.43±.79 | 42.40±.85 |
| | | BYOL | 87.64±.70 | 74.05±.84 | 35.62±.65 | 23.01±.43 | 58.12±.87 | 48.28±.88 | 38.23±.75 | 42.48±.82 |
| SL → MSL | 0→0.875 | SimCLR | 91.46±.66 | 77.62±.76 | 34.46±.64 | 22.50±.41 | 69.50±.87 | 51.27±.91 | 40.39±.82 | 62.12±.93 |
| | | BYOL | 88.37±.73 | 71.54±.78 | 36.08±.63 | **24.42**±.45 | 63.40±.86 | **53.65**±.88 | **46.62**±.85 | 64.33±.93 |
| MSL | 0 ↗ 1 | SimCLR | 81.32±.79 | 70.68±.82 | 32.70±.60 | 22.77±.41 | 61.36±.84 | 44.50±.83 | 36.27±.69 | 50.40±.86 |
| | | BYOL | 77.37±.83 | 67.84±.82 | 34.70±.64 | 23.38±.41 | 59.18±.82 | 45.37±.83 | 36.18±.71 | 51.00±.85 |

Table 11: 5-way 5-shot CD-FSL performance (%) of the models pre-trained with varying configurations of $\gamma$ in Eq. (3) of MSL. ResNet18 is used as the backbone model, and ImageNet is used as the source data for SL. The best results are marked in bold.

| Pre-train Scheme | $\gamma$ | Method | Small Similarity | | | | Large Similarity | | | |
| --- | --- | --- | --- | --- | --- | --- | --- | --- | --- | --- |
| | | | CropDisease | EuroSAT | ISIC | ChestX | Places | Plantae | Cars | CUB |
| SL | 0 | Default | 92.81±.45 | 84.73±.51 | 44.10±.58 | 25.51±.44 | 79.22±.64 | 63.21±.82 | 66.38±.80 | 83.93±.66 |
| SSL | 1 | SimCLR | 97.46±.34 | 94.12±.32 | 47.85±.65 | 25.26±.44 | 80.43±.61 | 60.07±.84 | 44.55±.74 | 47.36±.79 |
| | | BYOL | 96.93±.30 | 87.83±.48 | 47.59±.63 | 28.36±.46 | 72.47±.63 | 61.02±.82 | 48.56±.76 | 51.31±.78 |
| MSL | 0.875 | SimCLR | 96.50±.35 | 90.11±.40 | 45.38±.63 | 26.05±.44 | 82.56±.58 | 64.76±.83 | 51.84±.79 | 64.53±.80 |
| | | BYOL | 96.74±.31 | 90.82±.40 | 49.14±.70 | 29.58±.47 | 81.27±.59 | 67.39±.81 | 46.76±.73 | 69.67±.82 |
| SL → SSL | 0→1 | SimCLR | **97.88**±.30 | **95.28**±.27 | 48.38±.60 | 25.25±.44 | 84.40±.53 | 66.35±.82 | 51.31±.84 | 57.11±.88 |
| | | BYOL | 97.58±.26 | 91.82±.39 | 49.32±.63 | 28.27±.48 | 78.87±.60 | 67.83±.82 | 54.70±.84 | 60.60±.82 |
| SL → MSL | 0→0.875 | SimCLR | 97.49±.30 | 91.70±.35 | 47.43±.62 | 26.24±.44 | **85.76**±.52 | 69.24±.81 | 58.97±.82 | 81.51±.72 |
| | | BYOL | 97.09±.31 | 90.89±.40 | **50.72**±.67 | **30.20**±.48 | 83.29±.55 | **74.16**±.77 | **68.87**±.80 | **84.34**±.67 |
| MSL | 0 ↗ 1 | SimCLR | 94.83±.42 | 87.69±.50 | 44.48±.61 | 26.76±.45 | 80.62±.58 | 62.02±.83 | 52.97±.76 | 69.37±.79 |
| | | BYOL | 93.98±.41 | 86.66±.50 | 47.61±.66 | 28.55±.47 | 79.71±.60 | 63.77±.83 | 54.13±.75 | 70.49±.79 |

# K  Increasing Batch Size on the Source Dataset for MSL

Naturally, the size of labeled source data and unlabeled target data differ greatly. For example, while ImageNet contains 1.3M training examples, Cars and CUB each contains 3,400 and 2,350 unlabeled examples when 20% of the data is used. Thus, during 1,000 epochs of MSL pre-training (where each epoch corresponds to one pass through the unlabeled target data), only 2-3 passes are completed through ImageNet (refer to **MSL Pre-Training** setup in Appendix C.1). Considering the effectiveness of SL when domain similarity is large, we posit that MSL under large domain similarity can benefit from higher batch size for on the source data, *i.e.*, allowing more passes through the source dataset. In particular, we fix the batch size for the target data to 64, and increase the batch size for the source data.

Table 12 describes CD-FSL performance according to the source batch size on Cars and CUB. It is shown that larger batch size for the source dataset can improve the MSL performance. We suppose that the MSL model with ImageNet obtains large generalization ability from large-scale data, gaining much larger benefit than miniImageNet or tieredImageNet source. This improvement is significant in Cars and CUB datasets because they are similar to ImageNet. In fact, ImageNet data already includes car types ($\sim$10 classes) and bird species ($\sim$59 classes).

Table 12: 5-way $k$-shot performance of MSL according to the source batch size when ImageNet is used as source.

| Target Dataset | Method | Batch Size for Source Dataset | $k$=1 | $k$=5 |
|---|---|---|---|---|
| Cars | SimCLR | 64 (default) | 36.96$\pm$.77 | 51.84$\pm$.79 |
| | | 128 | 38.54$\pm$.81 | 53.80$\pm$.84 |
| | | 256 | 38.24$\pm$.78 | 54.18$\pm$.81 |
| | | 512 | 38.98$\pm$.81 | 55.25$\pm$.81 |
| | BYOL | 64 (default) | 33.31$\pm$.66 | 46.76$\pm$.73 |
| | | 128 | 39.85$\pm$.81 | 58.01$\pm$.80 |
| | | 256 | 41.45$\pm$.82 | 59.48$\pm$.80 |
| | | 512 | 40.98$\pm$.80 | 59.48$\pm$.81 |
| CUB | SimCLR | 64 (default) | 47.35$\pm$.87 | 64.53$\pm$.80 |
| | | 128 | 49.91$\pm$.87 | 68.01$\pm$.81 |
| | | 256 | 51.06$\pm$.85 | 69.51$\pm$.79 |
| | | 512 | 51.48$\pm$.88 | 70.13$\pm$.79 |
| | BYOL | 64 (default) | 50.71$\pm$.87 | 69.67$\pm$.82 |
| | | 128 | 52.75$\pm$.87 | 72.26$\pm$.79 |
| | | 256 | 54.17$\pm$.86 | 73.50$\pm$.79 |
| | | 512 | 53.70$\pm$.86 | 73.31$\pm$.80 |

# L Results Summary

## L.1 Source Dataset: ImageNet

Table 13 and Table 14 describe 5-way 1-shot and 5-way 5-shot CD-FSL performance when ImageNet is used as the source dataset, respectively. Note that Table 14 is added for convenience and this is the same with Table 3 in the main paper. The results of STARTUP on BSCD-FSL (*i.e.*, CropDisease, EuroSAT, ISIC, and ChestX) target datasets are from Phoo and Hariharan [47]. The results on the other four target datasets are our reimplementation with their official code.[7] Also, Islam et al. [30] did not provide the results of DynDistill on ImageNet source dataset, so we reimplemented it with their official code.[8]

Table 13: 5-way 1-shot CD-FSL performance (%) of the models pre-trained by SL, SSL, and MSL including their two-stage versions. ResNet18 is used as the backbone model, and ImageNet is used as the source data for SL. The balancing coefficient $\gamma$ in Eq. (3) of MSL is set to be 0.875. The best results are marked in bold and the second best are underlined.

| | Pre-train Scheme | Method | Small Similarity | | | | Large Similarity | | | |
|---|---|---|---|---|---|---|---|---|---|---|
| | | | CropDisease | EuroSAT | ISIC | ChestX | Places | Plantae | Cars | CUB |
| Single-Stage | SL | Default | 74.18±.82 | 66.14±.83 | 31.11±.55 | 22.48±.39 | 57.47±.86 | 43.66±.80 | **45.82**±.79 | **65.24**±.97 |
| | SSL | SimCLR | **91.00**±.76 | **84.30**±.73 | 36.39±.66 | 21.55±.41 | 64.97±.94 | 44.18±.85 | 32.46±.70 | 36.15±.76 |
| | | BYOL | 85.77±.73 | 66.16±.86 | 34.53±.62 | 22.75±.41 | 51.76±.79 | 42.16±.75 | 34.54±.70 | 36.50±.68 |
| | MSL | SimCLR | 88.38±.70 | 73.97±.79 | 34.02±.62 | 22.04±.40 | 65.13±.88 | 47.47±.86 | 36.96±.77 | 47.35±.87 |
| | | BYOL | 86.47±.74 | 73.18±.83 | **37.10**±.67 | **23.97**±.44 | 61.40±.87 | **48.31**±.86 | 33.31±.66 | 50.71±.87 |

(a) Performance comparison for single-stage schemes.

| | Pre-train Scheme | Method | CropDisease | EuroSAT | ISIC | ChestX | Places | Plantae | Cars | CUB |
|---|---|---|---|---|---|---|---|---|---|---|
| Two-Stage | SL→SSL | SimCLR | **92.24**±.70 | **86.51**±.67 | **36.11**±.67 | 21.75±.41 | **71.05**±.92 | 49.02±.91 | 37.43±.79 | 42.40±.85 |
| | | BYOL | 87.64±.70 | 74.05±.84 | 35.62±.65 | 23.01±.43 | 58.12±.87 | 48.28±.88 | 38.23±.75 | 42.48±.82 |
| | SL→MSL | SimCLR | 91.46±.66 | 77.62±.76 | 34.46±.64 | 22.50±.41 | 69.50±.87 | 51.27±.91 | 40.39±.82 | 62.12±.93 |
| | | BYOL | 88.37±.73 | 71.54±.78 | 36.08±.63 | 24.42±.45 | 63.40±.86 | 53.65±.88 | 46.62±.85 | 64.33±.93 |
| | SL→MSL+ | STARTUP | 85.10±.74 | 73.83±.77 | 31.69±.59 | 23.03±.42 | 66.02±.87 | 49.78±.93 | 45.75±.84 | **72.58**±.93 |
| | | DynDistill | 87.53±1.01 | 77.24±1.06 | 34.55±1.82 | 24.02±1.59 | 60.84±1.08 | 49.90±1.22 | 46.55±1.21 | 63.80±1.32 |

(b) Performance comparison for two-stage schemes.

Table 14: 5-way 5-shot CD-FSL performance (%) of the models pre-trained by SL, SSL, and MSL including their two-stage versions. ResNet18 is used as the backbone model, and ImageNet is used as the source data for SL. The balancing coefficient $\gamma$ in Eq. (3) of MSL is set to be 0.875. The best results are marked in bold and the second best are underlined.

| | Pre-train Scheme | Method | Small Similarity | | | | Large Similarity | | | |
|---|---|---|---|---|---|---|---|---|---|---|
| | | | CropDisease | EuroSAT | ISIC | ChestX | Places | Plantae | Cars | CUB |
| Single-Stage | SL | Default | 92.81±.45 | 84.73±.51 | 44.10±.58 | 25.51±.44 | 79.22±.64 | 63.21±.82 | **66.38**±.80 | **83.93**±.66 |
| | SSL | SimCLR | **97.46**±.34 | **94.12**±.32 | 47.85±.65 | 25.26±.44 | 80.43±.61 | 60.07±.84 | 44.55±.74 | 47.36±.79 |
| | | BYOL | 96.93±.30 | 87.83±.48 | 47.59±.63 | 28.36±.46 | 72.47±.63 | 61.02±.82 | 48.56±.76 | 51.31±.78 |
| | MSL | SimCLR | 96.50±.35 | 90.11±.40 | 45.38±.63 | 26.05±.44 | **82.56**±.58 | 64.76±.83 | 51.84±.79 | 64.53±.80 |
| | | BYOL | 96.74±.31 | 90.82±.40 | **49.14**±.70 | **29.58**±.47 | 81.27±.59 | **67.39**±.81 | 46.76±.73 | 69.67±.82 |

(a) Performance comparison for single-stage schemes.

| | Pre-train Scheme | Method | CropDisease | EuroSAT | ISIC | ChestX | Places | Plantae | Cars | CUB |
|---|---|---|---|---|---|---|---|---|---|---|
| Two-Stage | SL→SSL | SimCLR | **97.88**±.30 | **95.28**±.27 | 48.38±.60 | 25.25±.44 | 84.40±.53 | 66.35±.82 | 51.31±.84 | 57.11±.88 |
| | | BYOL | 97.58±.26 | 91.82±.39 | 49.32±.63 | 28.27±.48 | 78.87±.60 | 67.83±.82 | 54.70±.84 | 60.60±.82 |
| | SL→MSL | SimCLR | 97.49±.30 | 91.70±.35 | 47.43±.62 | 26.24±.44 | **85.76**±.52 | 69.24±.81 | 58.97±.82 | 81.51±.72 |
| | | BYOL | 97.09±.31 | 90.89±.40 | **50.72**±.67 | **30.20**±.48 | 83.29±.55 | 74.16±.77 | 68.87±.80 | 84.34±.67 |
| | SL→MSL+ | STARTUP | 96.06±.33 | 89.70±.41 | 46.02±.59 | 27.24±.46 | 85.00±.52 | 69.40±.84 | 68.43±.82 | **89.60**±.55 |
| | | DynDistill | 97.60±.35 | 92.28±.46 | 50.06±.86 | 29.65±.67 | 82.22±.81 | 71.49±1.06 | **69.45**±1.12 | 86.54±1.88 |

(b) Performance comparison for two-stage schemes.

---

[7] https://github.com/cpphoo/STARTUP
[8] https://github.com/asrafulashiq/dynamic-cdfsl

## L.2 Source Dataset: tieredImageNet

Table 15 and Table 16 describe 5-way 1-shot and 5-way 5-shot CD-FSL performance when tieredImageNet is used as the source dataset, respectively. Phoo and Hariharan [47] did not provide the results of STARTUP on tieredImageNet source dataset, so we reimplemented it with their official code. The results of DynDistill on BSCD-FSL are from Islam et al. [30]; however, note that DynDistill used a larger ResNet-18 backbone model than our setting, which is provided by Tian et al. [62]. Also, the results on the other four target datasets are our reimplementation with their official code.

The difference of the result of tieredImageNet from the result of ImageNet as the source dataset is that one-stage MSL can outperform SL on Cars and CUB datasets. It is considered that bigger source dataset makes SL stronger, as we have addressed this issue in Appendix I.

Table 15: 5-way 1-shot CD-FSL performance (%) of the models pre-trained by SL, SSL, and MSL including their two-stage versions. ResNet18 is used as the backbone model, and tieredImageNet is used as the source data for SL. The balancing coefficient $\gamma$ in Eq. (3) of MSL is set to be 0.875. The best results are marked in bold and the second best are underlined.

| | Pre-train Scheme | Method | Small Similarity | | | | Large Similarity | | | |
|---|---|---|---|---|---|---|---|---|---|---|
| | | | CropDisease | EuroSAT | ISIC | ChestX | Places | Plantae | Cars | CUB |
| Single-Stage | SL | Default | 65.70±.94 | 60.07±.88 | 29.75±.56 | 22.11±.42 | 52.82±.86 | 34.99±.64 | 31.38±.61 | 54.18±.91 |
| | SSL | SimCLR | **91.00**±.76 | **84.30**±.73 | 36.39±.66 | 21.55±.41 | 64.97±.94 | 44.18±.85 | 32.46±.70 | 36.15±.76 |
| | | BYOL | 85.77±.73 | 66.16±.86 | 34.53±.62 | 22.75±.41 | 51.76±.79 | 42.16±.75 | 34.54±.70 | 36.50±.68 |
| | MSL | SimCLR | 87.44±.72 | 77.42±.77 | 35.47±.64 | 21.95±.40 | 63.83±.93 | 46.47±.87 | 34.65±.74 | 50.41±.90 |
| | | BYOL | 84.67±.78 | 68.45±.81 | **37.30**±.66 | **24.41**±.44 | 60.07±.87 | **46.49**±.83 | **37.88**±.75 | **54.43**±.88 |
| | | | (a) Performance comparison for single-stage schemes. | | | | | | | |
| Two-Stage | SL→SSL | SimCLR | **92.41**±.70 | **86.61**±.66 | 36.95±.67 | 21.75±.40 | **68.51**±.94 | 47.92±.88 | 35.37±.77 | 44.74±.86 |
| | | BYOL | 84.82±.76 | 66.92±.84 | 37.19±.66 | 24.23±.46 | 44.34±.79 | 44.32±.81 | 38.49±.78 | 44.40±.83 |
| | SL→MSL | SimCLR | 90.13±.69 | 80.20±.78 | 35.32±.64 | 22.18±.38 | 64.85±.92 | 48.00±.86 | 35.83±.75 | 60.87±.90 |
| | | BYOL | 85.72±.76 | 53.92±.94 | **39.41**±.68 | 24.31±.45 | 59.16±.86 | **48.48**±.83 | **41.02**±.78 | 61.98±.88 |
| | SL→MSL+ | STARTUP | 77.67±.83 | 69.60±.86 | 33.90±.63 | 23.13±.40 | 59.14±.87 | 41.80±.85 | 34.45±.66 | **63.83**±.90 |
| | | DynDistill | 84.41±.75 | 72.15±.75 | 33.87±.56 | 22.70±.42 | 52.21±1.15 | 43.06±1.12 | 38.51±1.03 | 58.67±1.30 |
| | | | (b) Performance comparison for two-stage schemes. | | | | | | | |

Table 16: 5-way 5-shot CD-FSL performance (%) of the models pre-trained by SL, SSL, and MSL including their two-stage versions. ResNet18 is used as the backbone model, and tieredImageNet is used as the source data for SL. The balancing coefficient $\gamma$ in Eq. (3) of MSL is set to be 0.875. The best results are marked in bold and the second best are underlined.

| | Pre-train Scheme | Method | Small Similarity | | | | Large Similarity | | | |
|---|---|---|---|---|---|---|---|---|---|---|
| | | | CropDisease | EuroSAT | ISIC | ChestX | Places | Plantae | Cars | CUB |
| Single-Stage | SL | Default | 86.34±.60 | 79.95±.66 | 40.60±.58 | 25.20±.41 | 72.96±.67 | 51.11±.76 | 45.18±.68 | **74.14**±.80 |
| | SSL | SimCLR | **97.46**±.34 | **94.12**±.32 | 47.85±.65 | 25.26±.44 | 80.43±.61 | 60.07±.84 | 44.55±.74 | 47.36±.79 |
| | | BYOL | 96.93±.30 | 87.83±.48 | 47.59±.63 | 28.36±.46 | 72.47±.63 | 61.02±.82 | 48.56±.76 | 51.31±.78 |
| | MSL | SimCLR | 96.68±.33 | 91.72±.37 | 47.55±.67 | 26.10±.45 | **81.67**±.58 | 63.96±.82 | 48.81±.77 | 68.78±.82 |
| | | BYOL | 96.41±.33 | 89.51±.42 | **50.95**±.69 | **30.04**±.47 | 80.16±.60 | **67.09**±.80 | **54.75**±.80 | 73.03±.82 |
| | | | (a) Performance comparison for single-stage schemes. | | | | | | | |
| Two-Stage | SL→SSL | SimCLR | **97.88**±.31 | **95.39**±.26 | 50.28±.61 | 25.31±.44 | **83.51**±.56 | 65.40±.82 | 48.91±.83 | 61.80±.84 |
| | | BYOL | 96.25±.31 | 89.39±.45 | 53.00±.64 | 30.66±.48 | 71.57±.63 | 63.06±.79 | 55.04±.82 | 62.78±.80 |
| | SL→MSL | SimCLR | 97.43±.31 | 93.09±.33 | 49.66±.63 | 26.27±.44 | 83.03±.56 | 65.78±.85 | 52.22±.81 | 80.37±.76 |
| | | BYOL | 96.64±.32 | 85.97±.50 | **53.67**±.68 | **30.84**±.51 | 80.76±.56 | **69.77**±.79 | **62.09**±.78 | 82.77±.69 |
| | SL→MSL+ | STARTUP | 92.87±.41 | 85.23±.59 | 48.20±.62 | 27.06±.42 | 78.00±.60 | 60.28±.82 | 51.14±.75 | **83.36**±.66 |
| | | DynDistill | 95.90±.34 | 89.44±.42 | 47.21±.56 | 27.67±.46 | 75.67±.86 | 64.32±1.08 | 59.14±1.15 | 79.26±.97 |
| | | | (b) Performance comparison for two-stage schemes. | | | | | | | |

## L.3 Source Dataset: miniImageNet

Table 17 and Table 18 describe 5-way 1-shot and 5-way 5-shot CD-FSL performance when mini-ImageNet is used as the source dataset, respectively. The results of STARTUP and DynDistill on BSCD-FSL target datasets are from Phoo and Hariharan [47] and Islam et al. [30], respectively. The results on the other four target datasets are our reimplementation with their official codes. Similar to the results when tieredImageNet is used as the source dataset, one-stage MSL can outperform SL on Cars and CUB datasets. For a thorough comparison, we also report the results of meta-learning based approaches: MAML [17], MatchingNet [66], and RelationNet [58], where the numbers are from [64, 24, 30]. As previous studies on CD-FSL verified, meta-learning based algorithms are mostly outperformed by transfer learning based algorithms in the cross-domain setup.

Table 17: 5-way 1-shot CD-FSL performance (%) of the models pre-trained by SL, SSL, and MSL including their two-stage versions. ResNet10 is used as the backbone model, and miniImageNet is used as the source data for SL. The balancing coefficient $\gamma$ in Eq. (3) of MSL is set to be 0.875. The best results are marked in bold and the second best are underlined.

| Pre-train Scheme | Method | Small Similarity | | | | Large Similarity | | | |
|---|---|---|---|---|---|---|---|---|---|
| | | CropDisease | EuroSAT | ISIC | ChestX | Places | Plantae | Cars | CUB |
| **Single-Stage** | | | | | | | | | |
| - | MAML | - | - | - | - | - | - | - | - |
| | MatchingNet | 46.86±.88 | 54.88±.90 | 27.37±.51 | 20.65±.29 | 49.86±.79 | 32.70±.60 | 30.77±.47 | 35.89±.51 |
| | RelationNet | - | - | - | - | 48.64±.85 | 33.17±.64 | 29.11±.60 | 42.44±.77 |
| SL | Default | 72.82±.87 | 65.03±.88 | 29.91±.54 | 22.88±.42 | 52.45±.78 | 36.72±.67 | 30.20±.54 | 40.56±.78 |
| SSL | SimCLR | **89.49**±.74 | **79.50**±.78 | 34.90±.64 | 21.97±.41 | 58.75±.93 | 42.65±.80 | 30.89±.66 | 35.49±.73 |
| | BYOL | 80.10±.76 | 66.45±.80 | 33.50±.59 | 23.11±.42 | 47.81±.75 | 39.12±.71 | 31.53±.65 | 35.96±.70 |
| MSL | SimCLR | 87.15±.75 | 74.18±.80 | 35.10±.64 | 22.83±.41 | **59.72**±.89 | 42.24±.80 | 33.89±.66 | 40.89±.79 |
| | BYOL | 74.16±.82 | 66.64±.81 | **35.63**±.66 | **24.07**±.47 | 53.60±.82 | **43.94**±.79 | **35.71**±.68 | **42.73**±.78 |

(a) Performance comparison for single-stage schemes.

| Pre-train Scheme | Method | CropDisease | EuroSAT | ISIC | ChestX | Places | Plantae | Cars | CUB |
|---|---|---|---|---|---|---|---|---|---|
| **Two-Stage** | | | | | | | | | |
| SL→SSL | SimCLR | **89.39**±.82 | **82.64**±.73 | 35.09±.64 | 22.15±.40 | **63.19**±.92 | **46.30**±.85 | 34.85±.74 | 39.92±.79 |
| | BYOL | 82.61±.76 | 67.67±.77 | **35.92**±.68 | 23.76±.45 | 53.72±.79 | 45.02±.79 | 37.40±.74 | 41.61±.75 |
| SL→MSL | SimCLR | 86.18±.77 | 74.06±.85 | 33.91±.65 | 22.13±.40 | 61.56±.86 | 43.47±.79 | 35.78±.72 | 43.50±.82 |
| | BYOL | 75.77±.82 | 65.67±.83 | 35.23±.66 | **24.47**±.44 | 54.86±.81 | 44.68±.78 | **38.20**±.71 | **45.82**±.79 |
| SL→MSL+ | STARTUP | 75.93±.80 | 63.88±.84 | 32.66±.60 | 23.09±.43 | 48.87±.81 | 38.01±.73 | 31.79±.61 | 41.24±.75 |
| | DynDistill | 82.14±.78 | 73.14±.84 | 34.66±.58 | 23.38±.43 | 49.28±1.11 | 40.60±1.15 | 34.77±.98 | 42.51±1.11 |

(b) Performance comparison for two-stage schemes.

Table 18: 5-way 5-shot CD-FSL performance (%) of the models pre-trained by SL, SSL, and MSL including their two-stage versions. ResNet10 is used as the backbone model, and miniImageNet is used as the source data for SL. The balancing coefficient $\gamma$ in Eq. (3) of MSL is set to be 0.875. The best results are marked in bold and the second best are underlined.

| Pre-train Scheme | Method | Small Similarity | | | | Large Similarity | | | |
|---|---|---|---|---|---|---|---|---|---|
| | | CropDisease | EuroSAT | ISIC | ChestX | Places | Plantae | Cars | CUB |
| **Single-Stage** | | | | | | | | | |
| - | MAML | 78.05±.68 | 71.70±.72 | 40.13±.58 | 23.48±.96 | - | - | - | - |
| | MatchingNet | 66.39±.78 | 64.45±.63 | 36.74±.53 | 22.40±.70 | 63.16±.77 | 46.53±.68 | 38.99±.64 | 51.37±.77 |
| | RelationNet | 68.99±.75 | 61.31±.72 | 39.41±.58 | 22.96±.88 | 63.32±.76 | 44.00±.60 | 37.33±.68 | 57.77±.69 |
| SL | Default | 91.32±.49 | 84.00±.56 | 40.84±.56 | 27.01±.44 | 72.92±.66 | 53.26±.73 | 44.39±.66 | 58.10±.78 |
| SSL | SimCLR | **97.24**±.33 | **92.36**±.37 | 46.76±.61 | 25.62±.43 | 78.39±.61 | 59.77±.82 | 45.60±.72 | 47.69±.77 |
| | BYOL | 94.53±.41 | 86.55±.50 | 45.99±.63 | 27.71±.44 | 68.14±.68 | 55.31±.71 | 43.92±.70 | 49.34±.76 |
| MSL | SimCLR | 96.59±.35 | 90.34±.34 | **48.78**±.62 | 26.69±.44 | 78.17±.61 | 59.48±.82 | 48.36±.75 | 56.63±.78 |
| | BYOL | 93.71±.41 | 87.21±.48 | 48.63±.66 | **29.86**±.47 | 75.16±.64 | **63.45**±.81 | **53.33**±.76 | **60.66**±.77 |

(a) Performance comparison for single-stage schemes.

| Pre-train Scheme | Method | CropDisease | EuroSAT | ISIC | ChestX | Places | Plantae | Cars | CUB |
|---|---|---|---|---|---|---|---|---|---|
| **Two-Stage** | | | | | | | | | |
| SL→SSL | SimCLR | **96.84**±.40 | **94.51**±.32 | 48.23±.59 | 24.59±.42 | **81.52**±.56 | 64.37±.81 | 50.72±.80 | 55.06±.84 |
| | BYOL | 95.87±.35 | 90.01±.43 | **50.33**±.67 | 29.94±.48 | 75.83±.62 | 64.93±.79 | 55.46±.79 | 59.78±.81 |
| SL→MSL | SimCLR | 96.67±.33 | 90.18±.42 | 47.24±.63 | 26.47±.44 | 79.95±.61 | 61.34±.80 | 52.74±.79 | 61.33±.80 |
| | BYOL | 94.50±.39 | 87.96±.48 | 49.36±.66 | **30.23**±.50 | 76.67±.63 | **65.41**±.78 | **58.62**±.78 | **66.20**±.78 |
| SL→MSL+ | STARTUP | 93.02±.45 | 82.29±.60 | 47.22±.61 | 26.94±.44 | 69.56±.66 | 55.40±.78 | 46.73±.73 | 60.00±.78 |
| | DynDistill | 95.54±.38 | 89.07±.47 | 49.36±.59 | 28.31±.46 | 70.98±.94 | 58.63±1.14 | 51.98±1.18 | 62.86±1.06 |

(b) Performance comparison for two-stage schemes.

## L.4 Source Dataset: ImageNet (ResNet50)

Table 19 and Table 20 describe 5-way 1-shot and 5-way 5-shot CD-FSL performance when ResNet50 is used as a backbone and ImageNet is used as the source dataset, respectively. We find that the observations in our paper also hold for ResNet50.

Table 19: 5-way 1-shot CD-FSL performance (%) of the models pre-trained by SL, SSL, and MSL including their two-stage versions. ResNet50 is used as the backbone model, and ImageNet is used as the source data for SL. The balancing coefficient $\gamma$ in Eq. (3) of MSL is set to be 0.875. The best results are marked in bold and the second best are underlined.

| | Pre-train Scheme | Method | Small Similarity | | | | Large Similarity | | | |
|---|---|---|---|---|---|---|---|---|---|---|
| | | | CropDisease | EuroSAT | ISIC | ChestX | Places | Plantae | Cars | CUB |
| Single-Stage | SL | Default | $73.74_{\pm.90}$ | $67.38_{\pm.88}$ | $29.00_{\pm.51}$ | $22.03_{\pm.38}$ | $65.51_{\pm.91}$ | $46.52_{\pm.83}$ | $\mathbf{51.25}_{\pm.90}$ | $\mathbf{71.83}_{\pm.95}$ |
| | SSL | SimCLR | $\mathbf{90.40}_{\pm.77}$ | $\mathbf{81.82}_{\pm.77}$ | $35.26_{\pm.62}$ | $21.73_{\pm.41}$ | $63.98_{\pm.97}$ | $41.32_{\pm.80}$ | $32.91_{\pm.73}$ | $35.26_{\pm.75}$ |
| | | BYOL | $87.17_{\pm.73}$ | $72.71_{\pm.83}$ | $34.33_{\pm.63}$ | $22.67_{\pm.42}$ | $53.33_{\pm.84}$ | $39.34_{\pm.76}$ | $31.58_{\pm.71}$ | $33.38_{\pm.68}$ |
| | MSL | SimCLR | $87.17_{\pm.73}$ | $72.71_{\pm.83}$ | $34.33_{\pm.63}$ | $22.67_{\pm.42}$ | $\mathbf{68.24}_{\pm.90}$ | $45.85_{\pm.85}$ | $36.52_{\pm.77}$ | $47.53_{\pm.88}$ |
| | | BYOL | $87.25_{\pm.82}$ | $72.47_{\pm.84}$ | $\mathbf{36.68}_{\pm.67}$ | $\mathbf{23.54}_{\pm.43}$ | $62.75_{\pm.87}$ | $\mathbf{49.20}_{\pm.88}$ | $38.57_{\pm.77}$ | $48.72_{\pm.87}$ |

(a) Performance comparison for single-stage schemes.

| | Pre-train Scheme | Method | CropDisease | EuroSAT | ISIC | ChestX | Places | Plantae | Cars | CUB |
|---|---|---|---|---|---|---|---|---|---|---|
| Two-Stage | SL→SSL | SimCLR | $92.14_{\pm.72}$ | $\mathbf{86.41}_{\pm.65}$ | $36.31_{\pm.69}$ | $21.72_{\pm.41}$ | $70.58_{\pm.93}$ | $49.36_{\pm.91}$ | $37.49_{\pm.79}$ | $43.20_{\pm.87}$ |
| | | BYOL | $84.44_{\pm.97}$ | $69.11_{\pm.94}$ | $35.90_{\pm.69}$ | $21.62_{\pm.40}$ | $49.05_{\pm.89}$ | $37.40_{\pm.89}$ | $35.96_{\pm.75}$ | $36.95_{\pm.74}$ |
| | SL→MSL | SimCLR | $\mathbf{92.62}_{\pm.62}$ | $76.30_{\pm.79}$ | $35.51_{\pm.67}$ | $22.48_{\pm.42}$ | $\mathbf{73.05}_{\pm.88}$ | $54.08_{\pm.94}$ | $41.91_{\pm.85}$ | $61.51_{\pm.97}$ |
| | | BYOL | $91.04_{\pm.68}$ | $73.78_{\pm.81}$ | $\mathbf{37.27}_{\pm.67}$ | $\mathbf{24.70}_{\pm.42}$ | $65.55_{\pm.86}$ | $\mathbf{59.08}_{\pm.94}$ | $\mathbf{49.65}_{\pm.89}$ | $\mathbf{66.36}_{\pm.90}$ |

(b) Performance comparison for two-stage schemes.

Table 20: 5-way 5-shot CD-FSL performance (%) of the models pre-trained by SL, SSL, and MSL including their two-stage versions. ResNet50 is used as the backbone model, and ImageNet is used as the source data for SL. The balancing coefficient $\gamma$ in Eq. (3) of MSL is set to be 0.875. The best results are marked in bold and the second best are underlined.

| | Pre-train Scheme | Method | Small Similarity | | | | Large Similarity | | | |
|---|---|---|---|---|---|---|---|---|---|---|
| | | | CropDisease | EuroSAT | ISIC | ChestX | Places | Plantae | Cars | CUB |
| Single-Stage | SL | Default | $92.65_{\pm.47}$ | $85.95_{\pm.54}$ | $42.00_{\pm.60}$ | $25.04_{\pm.43}$ | $\mathbf{86.46}_{\pm.50}$ | $66.82_{\pm.80}$ | $\mathbf{73.49}_{\pm.80}$ | $\mathbf{91.06}_{\pm.54}$ |
| | SSL | SimCLR | $97.00_{\pm.37}$ | $\mathbf{93.34}_{\pm.34}$ | $46.16_{\pm.62}$ | $24.77_{\pm.43}$ | $79.02_{\pm.62}$ | $57.62_{\pm.85}$ | $43.97_{\pm.75}$ | $45.90_{\pm.77}$ |
| | | BYOL | $96.69_{\pm.30}$ | $86.19_{\pm.50}$ | $43.44_{\pm.61}$ | $27.21_{\pm.45}$ | $73.73_{\pm.65}$ | $58.40_{\pm.81}$ | $41.87_{\pm.71}$ | $48.33_{\pm.74}$ |
| | MSL | SimCLR | $96.83_{\pm.34}$ | $89.02_{\pm.46}$ | $45.49_{\pm.61}$ | $26.15_{\pm.44}$ | $84.97_{\pm.54}$ | $63.69_{\pm.83}$ | $51.11_{\pm.81}$ | $65.23_{\pm.80}$ |
| | | BYOL | $\mathbf{97.27}_{\pm.31}$ | $90.46_{\pm.41}$ | $\mathbf{49.06}_{\pm.69}$ | $\mathbf{28.98}_{\pm.47}$ | $82.19_{\pm.57}$ | $\mathbf{68.57}_{\pm.83}$ | $56.52_{\pm.81}$ | $66.99_{\pm.78}$ |

(a) Performance comparison for single-stage schemes.

| | Pre-train Scheme | Method | CropDisease | EuroSAT | ISIC | ChestX | Places | Plantae | Cars | CUB |
|---|---|---|---|---|---|---|---|---|---|---|
| Two-Stage | SL→SSL | SimCLR | $97.54_{\pm.34}$ | $\mathbf{95.57}_{\pm.27}$ | $48.95_{\pm.63}$ | $24.75_{\pm.44}$ | $85.39_{\pm.51}$ | $66.46_{\pm.85}$ | $50.54_{\pm.85}$ | $58.48_{\pm.89}$ |
| | | BYOL | $96.20_{\pm.38}$ | $91.94_{\pm.41}$ | $52.00_{\pm.66}$ | $26.42_{\pm.46}$ | $77.78_{\pm.61}$ | $66.95_{\pm.85}$ | $52.90_{\pm.81}$ | $54.37_{\pm.78}$ |
| | SL→MSL | SimCLR | $\mathbf{98.18}_{\pm.26}$ | $91.48_{\pm.38}$ | $49.34_{\pm.64}$ | $26.62_{\pm.45}$ | $\mathbf{87.87}_{\pm.49}$ | $72.39_{\pm.81}$ | $59.86_{\pm.85}$ | $81.46_{\pm.76}$ |
| | | BYOL | $97.80_{\pm.27}$ | $92.14_{\pm.35}$ | $\mathbf{53.04}_{\pm.67}$ | $\mathbf{30.91}_{\pm.51}$ | $85.72_{\pm.52}$ | $\mathbf{78.47}_{\pm.73}$ | $\mathbf{72.98}_{\pm.80}$ | $\mathbf{85.55}_{\pm.68}$ |

(b) Performance comparison for two-stage schemes.

## L.5 Source Dataset: ImageNet (20- and 50-shots)

Table 21 and Table 22 describe 5-way 20-shot and 5-way 50-shot CD-FSL performance when ResNet18 is used as a backbone and ImageNet is used as the source dataset, respectively. We find that the results are consistent with our main analysis. For target datasets that have small similarity to the source dataset, it remains beneficial to perform SSL pre-training on the unlabeled target data to adapt to target domain features, compared to SL on source (Obs. 4.1). For target datasets with large similarity, the relative benefit of SSL on target data is larger when few-shot difficulty is low (Obs. 5.2). We note that SL performance significantly benefits from large $k$ when similarity to the source domain is high. Furthermore, the observations about joint synergy via MSL (Obs. 6.1) and sequential synergy via two-stage pre-training (Obs. 6.2) consistently hold.

Table 21: 5-way 20-shot CD-FSL performance (%) of the models pre-trained by SL, SSL, and MSL including their two-stage versions. ResNet18 is used as the backbone model, and ImageNet is used as the source data for SL. The balancing coefficient $\gamma$ in Eq. (3) of MSL is set to be 0.875. The best results are marked in bold and the second best are underlined.

| | Pre-train Scheme | Method | Small Similarity | | | | Large Similarity | | | |
|---|---|---|---|---|---|---|---|---|---|---|
| | | | CropDisease | EuroSAT | ISIC | ChestX | Places | Plantae | Cars | CUB |
| Single-Stage | SL | Default | 97.21±.25 | 91.66±.33 | 53.78±.58 | 29.39±.42 | **89.65**±.40 | 76.08±.72 | **82.79**±.60 | 94.48±.38 |
| | SSL | SimCLR | 97.88±.29 | **96.02**±.23 | 55.94±.60 | 28.36±.43 | 83.90±.50 | 64.91±.80 | 49.64±.78 | 55.41±.77 |
| | | BYOL | 98.76±.15 | 94.60±.28 | 58.52±.55 | 35.26±.47 | 81.82±.52 | 72.32±.75 | 61.06±.79 | 64.38±.78 |
| | MSL | SimCLR | 98.78±.16 | 94.71±.25 | 56.39±.60 | 31.46±.43 | 88.64±.42 | 75.55±.73 | 66.05±.74 | 75.89±.68 |
| | | BYOL | **98.97**±.13 | 95.03±.24 | **60.54**±.62 | **37.35**±.49 | 88.06±.43 | **78.40**±.69 | 59.74±.72 | 80.23±.64 |

(a) Performance comparison for single-stage schemes.

| | Pre-train Scheme | Method | Small Similarity | | | | Large Similarity | | | |
|---|---|---|---|---|---|---|---|---|---|---|
| | | | CropDisease | EuroSAT | ISIC | ChestX | Places | Plantae | Cars | CUB |
| Two-Stage | SL→SSL | SimCLR | 98.27±.26 | **96.77**±.21 | 58.28±.57 | 28.57±.42 | 87.88±.42 | 71.52±.77 | 59.40±.82 | 67.79±.80 |
| | | BYOL | **99.20**±.12 | 96.60±.19 | 60.81±.60 | 36.00±.48 | 86.67±.43 | 78.91±.67 | 70.90±.76 | 73.95±.71 |
| | SL→MSL | SimCLR | 98.96±.15 | 95.58±.22 | 58.87±.59 | 31.78±.43 | **90.69**±.38 | 79.85±.67 | 76.54±.70 | 90.25±.48 |
| | | BYOL | 99.10±.14 | 95.82±.21 | **62.43**±.59 | **38.48**±.47 | 89.62±.40 | **84.00**±.61 | **84.39**±.60 | **91.59**±.46 |

(b) Performance comparison for two-stage schemes.

Table 22: 5-way 50-shot CD-FSL performance (%) of the models pre-trained by SL, SSL, and MSL including their two-stage versions. ResNet18 is used as the backbone model, and ImageNet is used as the source data for SL. The balancing coefficient $\gamma$ in Eq. (3) of MSL is set to be 0.875. The best results are marked in bold and the second best are underlined.

| | Pre-train Scheme | Method | Small Similarity | | | | Large Similarity | | | |
|---|---|---|---|---|---|---|---|---|---|---|
| | | | CropDisease | EuroSAT | ISIC | ChestX | Places | Plantae | Cars | CUB |
| Single-Stage | SL | Default | 98.19±.17 | 93.70±.27 | 60.42±.55 | 33.27±.44 | **91.49**±.34 | 80.96±.61 | **89.79**±.44 | **98.18**±.17 |
| | SSL | SimCLR | 97.99±.28 | **96.35**±.22 | 58.40±.56 | 30.61±.43 | 84.44±.47 | 66.60±.74 | 53.17±.71 | 62.66±.73 |
| | | BYOL | 99.04±.13 | 95.63±.24 | 62.74±.55 | 39.36±.49 | 84.26±.46 | 77.20±.68 | 70.37±.68 | 81.13±.57 |
| | MSL | SimCLR | 99.09±.13 | 95.93±.21 | 60.33±.57 | 35.52±.44 | 90.30±.35 | 80.41±.61 | 76.49±.66 | 89.97±.44 |
| | | BYOL | **99.27**±.10 | 96.00±.21 | **64.64**±.58 | **41.88**±.50 | 90.02±.36 | **82.68**±.60 | 70.34±.68 | 92.40±.37 |

(a) Performance comparison for single-stage schemes.

| | Pre-train Scheme | Method | Small Similarity | | | | Large Similarity | | | |
|---|---|---|---|---|---|---|---|---|---|---|
| | | | CropDisease | EuroSAT | ISIC | ChestX | Places | Plantae | Cars | CUB |
| Two-Stage | SL→SSL | SimCLR | 98.29±.26 | 97.04±.19 | 61.40±.54 | 30.67±.43 | 88.44±.39 | 73.16±.72 | 63.73±.77 | 75.53±.68 |
| | | BYOL | **99.44**±.09 | **97.35**±.16 | 66.71±.54 | 40.93±.50 | 88.74±.37 | 82.58±.60 | 81.17±.62 | 88.85±.45 |
| | SL→MSL | SimCLR | 99.34±.12 | 96.66±.18 | 64.44±.57 | 35.61±.45 | **92.09**±.33 | 83.88±.56 | 85.50±.55 | 96.82±.24 |
| | | BYOL | 99.37±.10 | 96.81±.18 | **67.90**±.57 | **43.59**±.49 | 91.36±.34 | **87.59**±.49 | **91.28**±.43 | 97.15±.23 |

(b) Performance comparison for two-stage schemes.

# M Additional Pre-training Schemes

## M.1 Alternative Two-Stage Schemes

In this section, we study alternative two-stage pre-training schemes. Namely, we consider MSL→SSL and SSL→MSL. By default, we use $\gamma = 0.875$ as the balancing hyperparameter for MSL within each scheme. However, for MSL→SSL, we also consider $\gamma = 0.125$ as a middle-ground between SL→SSL and MSL→SSL (with $\gamma = 0.875$).

Table 23 and Table 24 describe the few-show performances of the additional pre-training schemes (in the three lowermost rows), with other methods displayed for ease of comparison. We observe that MSL→SSL with either choice of $\gamma$ can achieve the best performance for target datasets with small similarity to the source dataset, e.g., CropDisease, ISIC, ChestX. On the other hand, SSL→MSL generally underperforms other methods, unable to achieve best performance for any of the target datasets. We posit that this is because the latter stage of pre-training (MSL) is closer to the source dataset compared to the former (SSL), thus learning undesirable source information before few-shot adaptation. For the second stage of MSL→SSL, we used the SGD optimizer for BYOL as well as SimCLR, to match the optimizer used in the first stage, MSL.

Table 23: 5-way 1-shot CD-FSL performance (%) of the models according to different two-stage pre-training schemes. ResNet18 is used as the backbone model, and ImageNet is used as the source data. If not specified otherwise, the balancing coefficient $\gamma$ in Eq. (3) of MSL is set to be 0.875. The best results are marked in bold.

| Pre-train Scheme | Method | Small Similarity | | | | Large Similarity | | | |
| --- | --- | --- | --- | --- | --- | --- | --- | --- | --- |
| | | CropDisease | EuroSAT | ISIC | ChestX | Places | Plantae | Cars | CUB |
| SL → SSL | SimCLR | 92.24±.70 | **86.51**±.67 | 36.11±.67 | 21.75±.41 | **71.05**±.92 | 49.02±.91 | 37.43±.79 | 42.40±.85 |
| | BYOL | 87.64±.70 | 74.05±.84 | 35.62±.65 | 23.01±.43 | 58.12±.87 | 48.28±.88 | 38.23±.75 | 42.48±.82 |
| SL → MSL | SimCLR | 91.46±.66 | 77.62±.76 | 34.46±.64 | 22.50±.41 | 69.50±.87 | 51.27±.91 | 40.39±.82 | 62.12±.93 |
| | BYOL | 88.37±.73 | 71.54±.78 | 36.08±.63 | 24.42±.45 | 63.40±.86 | **53.65**±.88 | **46.62**±.85 | **64.33**±.93 |
| MSL → SSL ($\gamma = 0.125$) | SimCLR | 92.18±.72 | 86.27±.68 | 36.20±.68 | 21.55±.41 | 67.12±.93 | 46.61±.88 | 34.86±.76 | 41.11±.82 |
| | BYOL | 84.33±.75 | 65.69±.83 | 36.13±.66 | 24.60±.44 | 48.03±.73 | 45.15±.78 | 40.62±.77 | 41.00±.76 |
| MSL → SSL ($\gamma = 0.875$) | SimCLR | **92.47**±.69 | 85.03±.69 | 36.22±.67 | 21.73±.41 | 67.74±.94 | 46.80±.86 | 35.08±.77 | 38.94±.80 |
| | BYOL | 92.09±.66 | 52.65±.90 | **37.51**±.65 | **24.73**±.44 | 46.13±.72 | 45.67±.81 | 38.80±.80 | 40.75±.76 |
| SSL → MSL | SimCLR | 84.42±.75 | 73.75±.82 | 33.80±.60 | 22.60±.42 | 62.63±.88 | 45.21±.82 | 38.51±.73 | 53.52±.89 |
| | BYOL | 78.26±.82 | 70.28±.78 | 35.37±.64 | 23.66±.41 | 60.88±.84 | 45.54±.84 | 40.27±.76 | 54.81±.86 |

Table 24: 5-way 5-shot CD-FSL performance (%) of the models according to different two-stage pre-training schemes. ResNet18 is used as the backbone model, and ImageNet is used as the source data. If not specified otherwise, the balancing coefficient $\gamma$ in Eq. (3) of MSL is set to be 0.875. The best results are marked in bold.

| Pre-train Scheme | Method | Small Similarity | | | | Large Similarity | | | |
| --- | --- | --- | --- | --- | --- | --- | --- | --- | --- |
| | | CropDisease | EuroSAT | ISIC | ChestX | Places | Plantae | Cars | CUB |
| SL → SSL | SimCLR | 97.88±.30 | **95.28**±.27 | 48.38±.60 | 25.25±.44 | 84.40±.53 | 66.35±.82 | 51.31±.84 | 57.11±.88 |
| | BYOL | 97.58±.26 | 91.82±.39 | 49.32±.63 | 28.27±.48 | 78.87±.60 | 67.83±.82 | 54.70±.84 | 60.60±.82 |
| SL → MSL | SimCLR | 97.49±.30 | 91.70±.35 | 47.43±.62 | 26.24±.44 | **85.76**±.52 | 69.24±.81 | 58.97±.82 | 81.51±.72 |
| | BYOL | 97.09±.31 | 90.89±.40 | 50.72±.67 | 30.20±.48 | 83.29±.55 | **74.16**±.77 | **68.87**±.80 | **84.34**±.67 |
| MSL → SSL ($\gamma = 0.125$) | SimCLR | 97.59±.34 | 95.26±.27 | 50.22±.60 | 24.68±.44 | 83.91±.53 | 64.91±.82 | 47.90±.81 | 56.02±.87 |
| | BYOL | 96.89±.30 | 88.96±.43 | 52.26±.65 | **31.19**±.51 | 72.16±.62 | 66.51±.77 | 58.57±.82 | 59.73±.80 |
| MSL → SSL ($\gamma = 0.875$) | SimCLR | 97.76±.32 | 94.87±.28 | 48.98±.61 | 24.92±.43 | 83.84±.54 | 64.35±.82 | 48.07±.82 | 52.65±.84 |
| | BYOL | **98.41**±.24 | 85.96±.49 | **53.22**±.66 | 30.67±.49 | 72.13±.65 | 67.51±.81 | 55.17±.82 | 59.08±.80 |
| SSL → MSL | SimCLR | 95.93±.37 | 89.86±.41 | 46.38±.63 | 26.69±.45 | 82.00±.58 | 63.38±.82 | 56.54±.79 | 73.12±.76 |
| | BYOL | 94.76±.38 | 88.76±.43 | 48.16±.66 | 29.23±.48 | 81.08±.58 | 64.52±.80 | 59.02±.79 | 74.61±.76 |

## M.2 Longer Training for SSL and MSL

Table 25 and Table 26 describe 5-way 1-shot and 5-way 5-shot CD-FSL performance of SSL and MSL when we increase the number of pre-training epochs from 1000 to 2000. We consider two schemes for this ablation: *Extended* and *Repeated*. In the *Extended* scheme, we simply adapt the milestones for the learning rate decay scheduler by doubling them from epochs 400, 600, 800 to 800, 1200, 1600. For the *Repeated* scheme, we apply the existing decay schedule in a cyclical manner; decaying the learning rate by a factor of 10 at epoch {400, 600, 800}, resetting the learning rate at epoch 1000, and again decaying at epoch {1400, 1600, 1800}. This scheme is used to isolate the effects of the learning rate reset that occurs during two-stage pre-training. Note that during two-stage pre-training, learning rate decay is applied to both stages independently, thus resulting in a jump in learning rate during the transition between stages. We follow the same implementation details as single-stage pre-training, unless explicitly stated.

We find that compared to standard single-stage pre-training, *Extended* pre-training for SSL achieves comparable 1-shot performance and minor overall improvement in 5-shot performance–up to 3.41% (Cars). On the other hand, MSL exhibits considerable performance increase overall under *Extended* pre-training. This is magnified under large domain similarity, where 5-shot performance improves by up to 18.02% (Cars). Considering this stark difference between SSL and MSL, we posit that longer training on MSL benefits from further extraction of source features, which are useful for similar target domains. However, we note that two-stage MSL still outperforms longer single-stage MSL, suggesting that SL pre-training is a more effective means of extracting source features. Comparing *Extended* and *Repeated* training, we find no major differences in performance, thus conclude that learning rate reset is not a major contributor in CD-FSL performance.

Table 25: 5-way 1-shot CD-FSL performance (%) of the models pre-trained by SSL and MSL under two schemes of longer pre-training: *Extended* and *Repeated*. ResNet18 is used as the backbone model, and ImageNet is used as the source data for SL. For the standard single-stage and two-stage pre-training, refer to Table 13 for comparison. The balancing coefficient $\gamma$ in Eq. (3) of MSL is set to be 0.875.

| | Pre-train Scheme | Method | Small Similarity | | | | Large Similarity | | | |
|---|---|---|---|---|---|---|---|---|---|---|
| | | | CropDisease | EuroSAT | ISIC | ChestX | Places | Plantae | Cars | CUB |
| Extended | SSL | SimCLR | 91.98±.76 | 86.34±.70 | 35.81±.67 | 21.44±.40 | 64.96±.95 | 43.81±.82 | 32.94±.72 | 35.92±.75 |
| | | BYOL | 87.13±.74 | 64.33±.81 | 35.50±.62 | 23.11±.42 | 50.46±.82 | 41.92±.75 | 35.78±.70 | 36.67±.68 |
| | MSL | SimCLR | 89.30±.69 | 74.73±.80 | 34.50±.60 | 22.32±.42 | 65.27±.88 | 48.95±.87 | 39.70±.80 | 54.88±.91 |
| | | BYOL | 90.90±.67 | 62.86±.86 | 36.22±.64 | 24.37±.42 | 50.94±.79 | 42.82±.76 | 44.35±.85 | 57.18±.91 |
| Repeated | SSL | SimCLR | 90.99±.77 | 85.96±.71 | 35.78±.64 | 21.70±.41 | 66.26±.96 | 44.53±.85 | 33.89±.74 | 36.86±.77 |
| | | BYOL | 86.36±.78 | 64.95±.80 | 35.15±.63 | 23.48±.43 | 50.21±.78 | 41.58±.75 | 35.73±.72 | 38.74±.70 |
| | MSL | SimCLR | 88.75±.69 | 73.47±.80 | 33.58±.61 | 22.32±.42 | 65.84±.88 | 48.64±.85 | 39.87±.79 | 54.44±.91 |
| | | BYOL | 89.70±.71 | 68.43±.82 | 36.76±.65 | 24.17±.44 | 53.47±.81 | 46.20±.80 | 43.80±.84 | 54.94±.84 |

Table 26: 5-way 5-shot CD-FSL performance (%) of the models pre-trained by SSL and MSL under two schemes of longer pre-training: *Extended* and *Repeated*. ResNet18 is used as the backbone model, and ImageNet is used as the source data for SL. For the standard single-stage and two-stage pre-training, refer to Table 14 for comparison. The balancing coefficient $\gamma$ in Eq. (3) of MSL is set to be 0.875.

| | Pre-train Scheme | Method | Small Similarity | | | | Large Similarity | | | |
|---|---|---|---|---|---|---|---|---|---|---|
| | | | CropDisease | EuroSAT | ISIC | ChestX | Places | Plantae | Cars | CUB |
| Extended | SSL | SimCLR | 97.44±.36 | 95.14±.28 | 49.47±.59 | 24.36±.42 | 82.21±.57 | 60.77±.85 | 45.30±.75 | 47.57±.80 |
| | | BYOL | 97.29±.28 | 89.54±.42 | 50.17±.64 | 29.16±.49 | 74.13±.65 | 62.91±.81 | 51.97±.78 | 53.35±.78 |
| | MSL | SimCLR | 97.39±.29 | 91.13±.37 | 47.28±.64 | 26.43±.44 | 84.32±.53 | 67.82±.82 | 56.97±.81 | 73.98±.79 |
| | | BYOL | 97.87±.26 | 89.27±.43 | 50.18±.67 | 30.82±.49 | 77.55±.60 | 66.32±.78 | 64.78±.81 | 77.18±.75 |
| Repeated | SSL | SimCLR | 97.33±.36 | 94.99±.28 | 49.43±.59 | 24.74±.43 | 82.54±.56 | 61.30±.84 | 46.21±.76 | 48.43±.81 |
| | | BYOL | 97.24±.27 | 89.92±.41 | 49.95±.63 | 29.64±.47 | 74.25±.63 | 62.86±.80 | 52.05±.79 | 55.73±.77 |
| | MSL | SimCLR | 97.12±.31 | 90.73±.38 | 46.23±.62 | 26.23±.43 | 83.94±.56 | 67.16±.81 | 56.88±.82 | 73.31±.81 |
| | | BYOL | 97.71±.27 | 86.99±.52 | 49.60±.69 | 30.37±.50 | 78.41±.59 | 69.56±.80 | 64.23±.82 | 75.00±.78 |

# N t-SNE Visualization of Pre-trained Models on the Target Domains

We provide t-SNE on the target datasets to visualize the difference of using the two types of pre-training models: supervised learning on the source domain and self-supervised learning on the target domain. Figure 14 describes t-SNE visualization of representations through SL/SSL models on the EuroSAT and CUB datasets. It is shown that on the EuroSAT dataset, representations through the SSL are clustered better than those through the SL; however, on the CUB dataset, representations through the SL are clustered better than those through the SSL. This implies that clustering ability of extractors trained through SL/SSL is related to the few-shot performance.

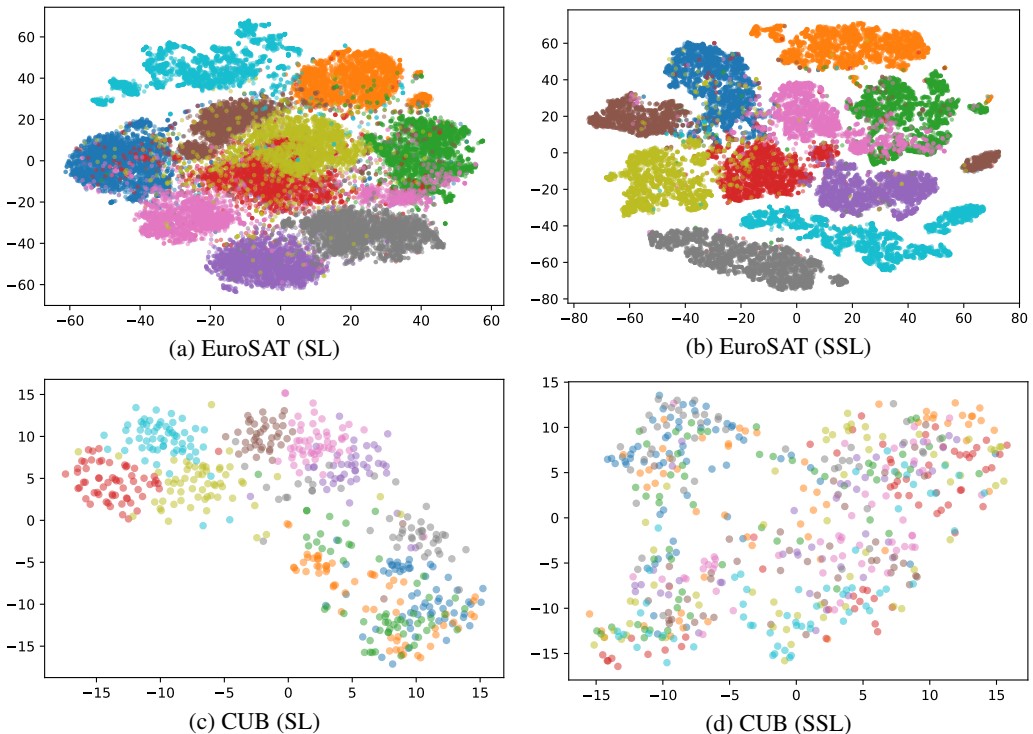

Figure 14: t-SNE visualization for each target dataset. ResNet18 is used as a backbone and ImageNet is used as the source dataset. Note that 10 classes are randomly sampled for the CUB dataset because CUB has 200 classes in total.

## O    Same Domain FSL Experiments

For the same domain FSL, although label space of the source and target datasets is still not shared, it is expected that SL is a better strategy and MSL improves the few-shot performance because MSL works like multi-task learning (MTL), improving the generalization ability. This is because the source and target datasets were collected in the same way.

To explain same-domain FSL experiments based on *domain similarity* and *few-shot difficulty*, we first provide them for the two datasets:

- Domain Similarity
    - miniImageNet $\leftrightarrow$ miniImageNet-test: 0.832
    - tieredImageNet $\leftrightarrow$ tieredImageNet-test: 0.869
- Few-shot Difficulty ($k = 5$)
    - miniImageNet-test: 0.467
    - tieredImageNet-test: 0.414

Interestingly, domain similarity of the miniImageNet-test dataset (0.832) is larger than that of every other benchmark, except for the Places (0.840). The Places dataset is found to be most similar to miniImageNet source. For the tieredImageNet-test dataset, domain similarity with tieredImageNet source (0.869) is larger than that of every other benchmark. Few-shot difficulty of miniImageNet-test is 0.467, which means slightly more difficult than Places and easier than Plantae. In addition, few-shot difficulty of tieredImageNet-test is 0.414, which means more difficult than EuroSAT and easier than Places.

Table 27 describes the few-shot performance under the same domain; miniImageNet $\rightarrow$ miniImageNet-test and tieredImageNet $\rightarrow$ tieredImageNet-test. As expected, they have large similarity and high difficulty. Therefore, (1) SL is more powerful than SSL, (2) MSL is a better strategy than both SL and SSL, and (3) two-stage pre-training boosts performance.

Table 27: 5-way $k$-shot FSL performance of the models pre-trained: miniImageNet $\rightarrow$ miniImageNet-test and tieredImageNet $\rightarrow$ tieredImageNet-test. We report the average accuracy and its 95% confidence interval over 600 few-shot episodes. B and S indicate base and strong augmentations, respectively. For MSL and SL$\rightarrow$MSL, $\gamma$ is set to 0.875.

| Pre-train Scheme | Method | Aug. | miniImageNet | | tieredImageNet | |
|---|---|---|---|---|---|---|
| | | | $k$=1 | $k$=5 | $k$=1 | $k$=5 |
| SL | Default | B | 54.89$\pm$.80 | 77.92$\pm$.59 | 60.98$\pm$.92 | 78.88$\pm$.68 |
| | | S | 57.30$\pm$.81 | 77.32$\pm$.65 | 60.77$\pm$.92 | 78.36$\pm$.71 |
| SSL | SimCLR | B | 42.69$\pm$.88 | 60.42$\pm$.81 | 51.63$\pm$.93 | 67.62$\pm$.84 |
| | | S | 54.39$\pm$.92 | 71.62$\pm$.79 | 66.67$\pm$1.02 | 80.60$\pm$.75 |
| | BYOL | B | 39.32$\pm$.76 | 58.36$\pm$.80 | 48.11$\pm$.88 | 69.72$\pm$.80 |
| | | S | 44.71$\pm$.80 | 63.66$\pm$.77 | 59.00$\pm$.97 | 78.59$\pm$.70 |
| MSL | SimCLR | S | 63.15$\pm$.85 | 80.03$\pm$.63 | 74.24$\pm$.94 | 86.90$\pm$.64 |
| | BYOL | S | 58.59$\pm$.84 | 78.17$\pm$.65 | 75.25$\pm$.92 | 88.37$\pm$.58 |
| SL$\rightarrow$SSL | SimCLR | S | 65.48$\pm$.89 | 83.84$\pm$.59 | 68.27$\pm$1.04 | 81.22$\pm$.75 |
| | BYOL | S | 55.76$\pm$.83 | 78.43$\pm$.61 | 65.78$\pm$.99 | 80.93$\pm$.66 |
| SL$\rightarrow$MSL | SimCLR | S | **67.24**$\pm$.86 | **85.02**$\pm$.52 | 74.14$\pm$.96 | 87.38$\pm$.62 |
| | BYOL | S | 61.10$\pm$.82 | 82.35$\pm$.58 | **75.47**$\pm$.90 | **88.72**$\pm$.58 |