# OpenReview forum: "Understanding Cross-Domain Few-Shot Learning Based on Domain Similarity and Few-Shot Difficulty"
_NeurIPS.cc/2022/Conference — NeurIPS 2022 Accept_

### Official Review · Reviewer_jPm7 · 2022-07-10

**Rating:** 5
**Confidence:** 4
**Soundness:** 3 good
**Presentation:** 3 good
**Contribution:** 3 good

**Summary:**

In this paper, the authors propose to use Domain Similarity and Few-Shot Difficulty to understand cross-domain few-shot learning. In particular, based on the Domain Similarity and Few-Shot Difficulty, the authors have several observations regarding the performance of self-supervised pre-training and supervised pre-training for cross-domain few-shot learning. Moreover, the authors also propose two additional pre-training strategies to improve the performance of cross-domain few-shot learning.

**Questions:**

1. Another experiment that is worth consideration is supervised pre-training on the labeled target domain data. This can further confirm the usefulness of self-supervised pre-training.

2. Would the observations hold even in the cases of larger shots, such as k=20 and k=50？

3. The analysis on the data augmentation is too coarse-grained. What's the individual impact of Gaussian blur and random gray scale on the performance of SSL?



**Limitations:**

Yes

**Strengths And Weaknesses:**

Strengths:
1. Overall, the paper is well-written and structured. The considered problem, i.e., cross-domain few-shot learning, is also important in the area of few-shot learning.
2. The paper also has several observations regarding cross-domain few-shot learning.
3. The experimental designs are interesting and shed light upon several things about cross-domain few-shot learning.

Weaknesses:

1. Most of the observations are not surprising. Due to the domain difference, it is expected that self-supervised pre-training on the target can achieve better results than supervised pre-training on the source.

2. The domain similarity in the paper is computed based on EMD. It is not clear if the computed domain similarity reflects the actual similarity between the datasets.

3. The lack of visualization on the trained representation makes it hard to understand what causes the difference between self-supervised pre-training and supervised pre-training for cross-domain few-shot learning.

4. The proposed improved schemes achieve marginal performance improvement compared with the baselines.

######################
Post-rebuttal: the rebuttal provided by the authors addresses my concerns. In the revised version, the authors are expected to add the additional results which can greatly improve the paper. I increase my score from 4 to 5.

---

> ### Author Response · Authors · 2022-08-02
> **Response to Reviewer jPm7 (2/2)**
>
> ### W4. Marginal improvements.
>
> -   [Please refer to Lines 318-321] Also in Section 6, our main goal is the analysis of basic pre-training methods for CD-FSL. Thus, we investigated the synergy when SL and SSL are jointly used (Observation 6.1) and confirmed whether the two-stage pre-training scheme (Observation 6.2) used in recent works [1,2] is useful.
> -   [Please refer to Lines 352-354] As a result, our simple two-stage methods, without any additional techniques, are shown to consistently improve performance compared to the single-stage counterparts. They also achieve comparable performance to the meticulously designed two-stage approaches such as STARTUP [1] and DynDistill [2], even though our primary goal is not performance improvement. Our findings with the improved schemes reveal the boosting factor of (1) joint usage and (2) sequential usage of SL and SSL in pre-training for CD-FSL, which can be a hint for future work.
> -   Note that STARTUP [1] trains a student model with knowledge distillation from a teacher model, thus employing two networks in hand and optimizing three training loss functions simultaneously. DynDistill [2] uses two types of augmentation and also a teacher-student structure.
>
>
>
>
>
> &nbsp;
>
>
>
> ### Q1. Supervised pre-training on the target domain.
>
>
>
> -   [Please refer to Lines 186-187] We used supervised pre-training on the target domain to define few-shot difficulty by an empirical upper bound of the few-shot performance.
> -   [Please refer to Appendix E] The results of supervised pre-training on the target domain are also provided. It is natural that supervised pre-training on the target domain overwhelms self-supervised pre-training on the target domain. However, the premise of few-shot learning is scarcity of labeled data for the target domain, and our setting attempts to tackle this challenge by exploiting unlabeled data in the target domain. Therefore, we focus on "SL on the source domain vs. SSL on the target domain" for this work.
>
>
>
>
> &nbsp;
>
>
>
> ### Q2. 20- and 50-shot results.
>
>
>
> [Please refer to Appendix O.4 in the revised manuscript] We added the results of 20- and 50-shots. We find that the results are consistent with our main analysis:
> -   For target datasets that have small similarity to the source dataset, it remains beneficial to perform SSL pre-training on the unlabeled target data to adapt to target domain features, compared to SL on source (Observation 4.1).
> -   For target datasets with large similarity, the relative benefit of SSL on target data is larger when few-shot difficulty is low (Observation 5.2). We note that SL performance significantly benefits from large $k$ when similarity to the source domain is high.
> -   Furthermore, the observations about joint synergy via MSL (Observation 6.1) and sequential synergy via two-stage pre-training (Observation 6.2) consistently hold.
>
>
>
>
> &nbsp;
>
>
>
> ### Q3. Data augmentation ablation study.
>
>
>
> -   Our main goal about data augmentation experiments is to show that few-shot performance varies to augmentation techniques when self-supervised learning is used for pre-training. Therefore, the detailed performance and effectiveness according to each augmentation technique is beyond our scope. However, it can be useful information for CD-FSL so that we will update if time allows.
>
>
>
>
> &nbsp;
>
>
>
> [1] Phoo, Cheng Perng, and Bharath Hariharan. "Self-training for few-shot transfer across extreme task differences." International Conference on Learning Representations. 2021.
> [2] Islam, Ashraful, et al. "Dynamic distillation network for cross-domain few-shot recognition with unlabeled data." Advances in Neural Information Processing Systems 34 (2021): 3584-3595.
> [3] Cui, Yin, et al. "Large scale fine-grained categorization and domain-specific transfer learning." Proceedings of the IEEE conference on computer vision and pattern recognition. 2018.
> [4] Li, Hao, et al. "Rethinking the hyperparameters for fine-tuning." International Conference on Learning Representations. 2020.

---

> ### Author Response · Authors · 2022-08-02
> **Response to Reviewer jPm7 (1/2)**
>
> Thank you for your thoughtful comments. We attached the revised manuscript to **the end of the supplementary file** (Appendix O), taking the reviewers’ comments into deep consideration.
>
>
>
> Before addressing each weakness and question, we would like to recap the setup, and main contribution of our paper. We address the real-world CD-FSL setting in which unlabeled target data is available during pre-training. In this setting, our contribution starts with a novel observation that standard self-supervised learning (SSL), using off-the-shelf methods, can be an effective way to utilize this unlabeled target data in CD-FSL (Observations 4.1 and 4.2). This approach has been dismissed in previous literature on account of the small scale of unlabeled data [1,2], however, our findings show that it is not only viable, but highly effective. Thus, we identify two distinct pre-training strategies for CD-FSL: supervised learning (SL) on **labeled source** data and self-supervised learning (SSL) on **unlabeled target** data. We perform foundational comparison and analysis of these two strategies under various datasets and methods, and provide insights on when they are most effective (Observations 5.1 and 5.2) and how to effectively take advantage of both strategies (Observations 6.1 and 6.2).
>
>
>
> &nbsp;
>
>
>
> We itemize the weaknesses and questions you mentioned and answer them below.
>
>
>
> ### W1. Novelty.
> -   [Please refer to Lines 46-53, 238-240, Observations 4.1 and 4.2] Our observations are in direct opposition to previous works [1,2]. Previous works said SSL will not work because unlabeled target data is too small. However, we show that SSL with strong augmentation on the target domain can achieve better results than SL on the source domain, even when the unlabeled target is much smaller than the source (1.3M (source) vs. 2K-8K (target)).
> -   [Please refer to Lines 279-282, Observations 5.1 and 5.2] Furthermore, our main discussion on CD-FSL is not only restricted to domain similarity, but we discovered that few-shot difficulty must be considered in tandem. For instance, SSL exhibits higher accuracy than SL on the Places dataset with the largest domain similarity, but SL exhibits higher accuracy than SSL on the CUB dataset with the second largest domain similarity. We explored why this inconsistency occurs by taking few-shot difficulty into account. Our observations provide a comprehensive analysis based on both domain similarity and few-shot difficulty.
>
>
>
>
> &nbsp;
>
>
>
> ### W2. EMD as a domain similarity measure.
>
> -   [Please refer to Appendix D] To measure domain similarity, we must estimate it because there is **no ground-truth** domain similarity. Thus, Earth Moving Distance (EMD) can be a good estimation because the distance between the two domains can be considered as the cost of moving images from one domain to the other. Previous studies in the transfer learning context [3,4] have also used EMD to estimate the domain similarity or domain difference. In addition, EMD has advantages compared to other metric choices, e.g., Kullback-Leibler divergence (KLD), Jensen-Shannon divergence (JSD), and maximum mean discrepancy (MMD). This is because EMD can be computed directly from the samples and EMD is robust to hyperparameters.
> -   Furthermore, we divided all target datasets into two groups based on EMD: large similarity and small similarity groups. The datasets in the small similarity group are BSCD-FSL datasets. This result implies that the domain similarity we used reflects the actual similarity. This is because BSCD-FSL benchmarks are designed for cross-domain by keeping the target domains away from the source domain. We can visually observe that BSCD-FSL datasets are far from the source dataset, looking at the examples displayed in Appendix B.2.
>
>
>
>
> &nbsp;
>
>
>
> ### W3. The lack of visualization.
>
> -   [Please refer to Appendix O.3 in the revised manuscript] We added t-SNE on EuroSAT and CUB datasets using the two types of pre-training models: (1) supervised learning on the source and (2) self-supervised learning on the target. It is shown that on the EuroSAT dataset, representations through the SSL are clustered better than those through the SL; however, on the CUB dataset, representations through the SL are clustered better than those through the SSL. This implies that the clustering ability of extractors trained through SL/SSL is related to the performance provided in Table 3.

---

### Official Review · Reviewer_eHxj · 2022-07-12

**Rating:** 4
**Confidence:** 3
**Soundness:** 1 poor
**Presentation:** 3 good
**Contribution:** 1 poor

**Summary:**

This paper investigates using (extra) unlabeled target domain data for cross-domain few-shot learning (CD-FSL), as well as the impact of source-target domain similarity and the inherent few-shot difficulty. Specifically, given the unlabeled target domain data, the authors set up two learning schemes, i.e., the self-supervised learning and the mixed-supervised learning. Through extensive experiments, they reveal some observations w.r.t. the domain similarity and the few-shot difficulty.

**Questions:**


- Do the unlabeled data and the labeled data have overlapping classes? How do you ensure this?
- In practice, how do you identify the target domain in advance and prepare the unlabeled images?
- Why do you use Eq.5 to measure the difficulty of few-shot learning? You should explain its rationality. If you choose different settings of $\beta$ and $k$, does it change your observations?

**Limitations:**

The authors have pointed out some limitations in the conclusion part.

**Strengths And Weaknesses:**

### Strengths
- Extensive experiments.
- The paper is easy to follow.

### Weakness

- The assumption that there are extra unlabeled samples for the target domain is controversial. In realistic few-shot learning, the testing set (consisted of the support and query images) may be collected on-the-fly. How can you identify the target domain in advance and collect the unlabeled images?
- In Section 4, some important descriptions are misleading as if SSL is superior than SL for base-training (e.g., L224 "investigating the superiority of SSL over SL for pre-training"). The real reason is that they use different domains for base-training: the SSL uses the target-domain and the SL uses the source domain.
- It is not clear that on the target domain, whether the unlabeled data (for base-training) and the labeled data (for support and query) have overlapping classes.
- In overall, the observations and discoveries provide limited knowledge for understanding the cross-domain few-shot learning. Adding target domain data improves target-domain accuracy is natural, even if the target domain samples are unlabeled.

---

> ### Author Response · Authors · 2022-08-02
> **Response to Reviewer eHxj (2/2)**
>
>
> ### W3 & Q1. Overlapped classes between pre-training and fine-tuning/evaluation.
>
> -   As we answered for (W1 & Q2), it is plausible to identify the target domain in advance. In our setting, we sample an unlabeled dataset $\mathcal{D}_U$ from the entire target dataset $\mathcal{D}_N$ randomly. Therefore, the classes of an unlabeled dataset (for pre-training) $\mathcal{D}_U$ and a labeled dataset (for fine-tuning/evaluation) $\mathcal{D}_L$ are overlapped. Note that label information is *not* used for the SSL pre-training, although the label distributions overlap between pre-training and fine-tuning/evaluation sets.
> -   For real-world scenarios, e.g., when unlabeled images are crawled from the internet, class overlapping may or may not occur. However, even if the classes are not overlapped, we can still collect the unlabeled data related to the labeled dataset.
>
>
> &nbsp;
>
> ### W4. Adding target domain data naturally improves target domain accuracy.
>
> -   The main purpose of our analysis is *not* adding target domain data to improve the target accuracy. Instead, we would like to find a better strategy between SL on source and SSL on target because it is not always the case that "adding target domain data naturally improves target domain accuracy." Our extensive experiments with various scenarios revealed that the better strategy differs according to domain similarity and few-shot difficulty.
> -   [Please refer to Table 3] In detail, we found that *SL on the source domain* is a better strategy for the dataset whose domain similarity is large and few-shot difficulty is high. As can be seen in Table 3, the CUB dataset has SL performance of 83.93%, while SSL and MSL (which use target domain data) performances are only 51.31% and 69.67%, respectively, because the CUB dataset has large domain similarity with high few-shot difficulty.
>
>
> &nbsp;
>
> ### Q3. Few-shot difficulty measure.
>
> -   [Please refer to Lines 184-190] We proposed few-shot difficulty to quantify the difficulty of the target dataset itself, regardless of its relationship to the source dataset. Therefore, we measured the empirical upper bound of few-shot performance, and the difficulty can be defined as a monotonically decreasing function of the performance. To measure the empirical upper bound, we used the subset of the target dataset as labeled data to pre-train the model and evaluated on the remaining unseen target data, because the generalization capability indicates the hardness. This shares a similar idea to the experimental setting in [8] that used full supervision of unlabeled target data for the upper bound of semi-supervised few-shot learning.
> -   Furthermore, $\beta$ is irrelevant to the relative order of few-shot difficulty because it is only a scaling parameter inside the exponential.
> -   [Please refer to Appendix E] We showed that the order of few-shot difficulty is consistent across various choices of $k$.
>
> &nbsp;
>
> [1] Phoo, Cheng Perng, and Bharath Hariharan. "Self-training for few-shot transfer across extreme task differences." International Conference on Learning Representations. 2021.
> [2] Islam, Ashraful, et al. "Dynamic distillation network for cross-domain few-shot recognition with unlabeled data." Advances in Neural Information Processing Systems 34 (2021): 3584-3595.
> [3] Comer, Joseph F., Philip L. Jacobson, and Heiko Hoffmann. "Few-Shot Image Classification Along Sparse Graphs." Proceedings of the IEEE/CVF Conference on Computer Vision and Pattern Recognition. 2022.
> [4] Ganin, Yaroslav, and Victor Lempitsky. "Unsupervised domain adaptation by backpropagation." International conference on machine learning. PMLR, 2015.
> [5] Long, Mingsheng, et al. "Unsupervised domain adaptation with residual transfer networks." Advances in neural information processing systems 29 (2016).
> [6] Wang, Rui, et al. "Cross-domain contrastive learning for unsupervised domain adaptation." IEEE Transactions on Multimedia (2022).
> [7] Shen, Kendrick, et al. "Connect, not collapse: Explaining contrastive learning for unsupervised domain adaptation." International Conference on Machine Learning. PMLR, 2022.
> [8] Li, Xinzhe, et al. "Learning to self-train for semi-supervised few-shot classification." Advances in Neural Information Processing Systems 32 (2019).

---

> > ### Comment · Reviewer_eHxj · 2022-08-09
> > **I still have concerns on the setting of using unlabeled target-domain samples**
> >
> > 1） Some prior explorations on this setting do not endorse the value of this setting. If you insist that in realistic few-shot learning, the unlabeled target-domain samples are accessible in advance, you should at least compare your setting against the semi-supervised setting, \emph{i.e.}, using extra unlabeled sampled during the fine-tuning stage. Which setting would be more beneficial for the few-shot accuracy?
> > 2) Using 20% testing data as the unlabeled target-domain samples actually brings a very large data increase, because the total testing data volume is much larger than the support samples in a few-shot regime (e.g., 5-way 1-shot only have 5 support samples, while 20% testing data is much more than 5).

---

> > > ### Author Response · Authors · 2022-08-09
> > > **Further response to Reviewer eHxj**
> > >
> > > Thank you for sharing your concerns. We are encouraged to address each of them as following.
> > >
> > > We briefly summarize our opinions about the setting of using unlabeled target-domain samples. As we have responded, there are real-world cases where unlabeled target data is available such as X-ray or satellite images. In these situations, we do not argue that "using target dataset always improve performances." Rather, we compare SL on source and SSL on target in the pre-training phase for CD-FSL.
> > >
> > > ### (1) Discussion on the semi-supervised few-shot learning setting.
> > >
> > > - Regarding the problem setting of using unlabeled target-domain samples, while we do refer to some prior works, we have also pointed out many examples of real-world scenarios in which the availability of some unlabeled target data is plausible.
> > > - In this problem setup, we focus our analysis on pre-training schemes for CD-FSL (please refer to lines 141-142), thus we evaluate each pre-training scheme *under a fixed fine-tuning phase*. This is because there is no in-depth study on comparison between SL on source and SSL on target in the pre-training phase.
> > > - Semi-supervised few-shot learning (SS-FSL) has been proposed to use unlabeled data during **fine-tuning**, while assuming that representations trained on the source dataset are good enough. Unlike SS-FSL, our study focuses on which **pre-training** scheme learns better representations for few-shot learning.
> > > - Although semi-supervised learning can be applied in the fine-tuning phase, it requires advanced techniques that fit with semi-supervised setting. Therefore, we believe that comparing the basic pre-training schemes against the semi-supervised setting is beyond the scope of our analysis. We note that SS-FSL is also a valuable research direction, and our pre-training schemes can be combined with SS-FSL methods for further improvement of CD-FSL.
> > >
> > > ### (2) Large volume of 20% target data.
> > >
> > > - As you mentioned, 20% of target data is larger than the few-shot support set. However, unlike support samples (i.e., labeled few-shot samples), this is in unlabeled form, which is available in many few-shot scenarios where large-scale annotation is prohibitive, but sample collection is not, such as X-ray and satellite images.
> > > - Furthermore, our analysis focuses on pre-training schemes for CD-FSL. Compared to SL pre-training on source datasets such as ImageNet ($\sim$1.3M images), unlabeled target data used for SSL pre-training on target datasets is rather small, to the tune of thousands of images.
> > > - In addition, we note that SSL pre-training can achieve remarkable results even when very small scale unlabeled data (e.g., 5%, or hundreds of samples) is used. Please refer to Appendix G.
> > >
> > > We hope we have addressed your concerns. Please let us know if there is anything else you would like for us to address.

---

> ### Author Response · Authors · 2022-08-02
> **Response to Reviewer eHxj (1/2)**
>
> Thank you for your thoughtful comments. We have updated the manuscript to **the supplementary file**, taking the reviewers’ comments into deep consideration.
>
> Before addressing each weakness and question, we would like to recap the setup, and main contribution of our paper. We address the real-world CD-FSL setting in which *unlabeled target data is available* during pre-training. This setting has recently gained attention [1,2] as a plausible way to tackle the large differences between source and target domains in CD-FSL, because many FSL problem domains already have unlabeled data such as undiagnosed X-ray images, unlabeled satellite images [1], along with the option to collect additional data via web crawling.
>
> In this setting, our contribution starts with a novel observation that standard self-supervised learning (SSL), using off-the-shelf methods, can be an effective way to utilize this unlabeled target data in CD-FSL (Observations 4.1 and 4.2). This approach has been dismissed in previous literature on account of the small scale of unlabeled data [1,2], however, our findings show that it is not only viable, but highly effective. Thus, we identify two distinct pre-training strategies for CD-FSL: supervised learning (SL) on **labeled source** data and self-supervised learning (SSL) on **unlabeled target** data. We perform foundational comparison and analysis of these two strategies under various datasets and methods, and provide insights on when they are most effective (Observations 5.1 and 5.2) and how to effectively take advantage of both strategies (Observations 6.1 and 6.2).
>
> &nbsp;
>
> We itemize the weaknesses and questions you mentioned and answer them below.
>
>
>
> ### W1 & Q2. Justification on the "unlabeled target" settings.
>
> - This setting is in line with many prior works [1,2,3], which developed their models under the assumption that unlabeled target data is accessible. Not only in few-shot learning, but it is also a widely studied setting in unsupervised domain adaptation literature [4,5,6,7]. Furthermore, it is possible to identify the target domain in advance, which is plausible in practice because goal setting takes precedence over model development in real-world scenarios.
> - For example, let us say that we want to develop a model to classify X-ray images with few labeled samples. To mitigate the difficulty of training deep models using a few labeled samples from the target domain, we can consider two pre-training strategies.
> 	1. We may use publicly available large-scale datasets such as ImageNet to pre-train the model on a large number of labeled samples from a more general **source domain**.
> 	2. Alternatively, we may exploit unlabeled data pertaining to the **target domain**. This unlabeled data can be obtained through web-crawling, or we may use stored X-ray images that are readily available yet unlabeled.
> - Note that we do not address streaming/online settings involving on-the-fly data collection. In our setting, we assume we have enough time to collect unlabeled data in advance, such as in the aforementioned example scenario regarding medical imagery. This assumption follows that of previous work on CD-FSL [1,2,3].
>
> &nbsp;
>
> ### W2. Line 224 is misleading.
>
> -   [Please refer to Line 224 in the revised manuscript] We clarified this sentence as: "investigating the superiority of SSL over SL for pre-training" $\rightarrow$ "investigating the superiority of SSL **on target** over SL **on source** for pre-training."

---

### Official Review · Reviewer_Ger5 · 2022-07-19

**Rating:** 5
**Confidence:** 4
**Soundness:** 3 good
**Presentation:** 4 excellent
**Contribution:** 3 good

**Summary:**

This paper studies the problem of cross-domain few-shot learning in images, and focuses on pre-training and fine-tuning schemes. Through empirical study, the paper shows the findings about the effectiveness of supervised pre-training on base classes and self-supervised pre-training on few-shot data. The paper also incorporates two metrics, domain similarity and few-shot difficulty, to help diagnose the model performance. Finally, a two-stage pre-training scheme with a combination of supervised learning and self-supervised supervised learning is proposed to improve performance in cross-domain few-shot learning.



**Questions:**

Please refer to the above "Weaknesses".

**Limitations:**

No limitation and potential negative societal impact section is included in the main paper.


**Strengths And Weaknesses:**

- Strengths:
This is one experimental analysis paper, and the paper is organized in a systematic way in both writing and experimental presentation.

- Weaknesses:
1. To quantify the difficulty of a dataset, the paper defines the metric of few-shot difficulty, based on the empirical upper bound of few-shot performance. Specifically, it is shown in L188 that “we use 20% of the target dataset as labeled data to pre-train the model in a supervised manner. Then, the pre-trained model is evaluated on the remaining unseen target data for the 5-way k-shot classification task.” One question arises for the selection of the 20% of the target dataset as labeled data, is there a class split control? the class split control will affect the baseline results significantly which is already shown in some few-shot related papers. Also, the random selection for constructing the “5-way k-shot classification task” will also affect the baseline results. How are the class split control and the construction of “5-way k-shot classification task” realized?

2. L218: In implementation, though different backbone networks (e.g. ResNet18 and ResNet10) are used depending on the source data, they are still shallow backbones. Will these conclusions still hold for deeper backbones?

---

> ### Author Response · Authors · 2022-08-02
> **Response to Reviewer Ger5**
>
> Thank you for your thoughtful comments.
> We attached the revised manuscript to **the end of the supplementary file** (Appendix O), taking the reviewers’ comments into deep consideration.
>
> Before addressing each weakness and question, we would like to recap the setup, and main contribution of our paper. We address the real-world CD-FSL setting in which unlabeled target data is available during pre-training. In this setting, our contribution starts with a novel observation that standard self-supervised learning (SSL), using off-the-shelf methods, can be an effective way to utilize this unlabeled target data in CD-FSL (Observations 4.1 and 4.2), contrary to assumptions in previous works [1, 2]. Thus, we identify two distinct pre-training strategies for CD-FSL: supervised learning (SL) on **labeled source** data and self-supervised learning (SSL) on **unlabeled target** data. We perform foundational comparison and analysis of these two strategies under various datasets and methods, and provide insights on when they are most effective (Observations 5.1 and 5.2) and how to effectively take advantage of both strategies (Observations 6.1 and 6.2).
>
> &nbsp;
>
> We itemize the weaknesses and questions you mentioned and answer them below.
>
> ### W1. For few-shot difficulty, 20% split of the target dataset and 5-way $k$-shot construction.
>
> -   [Please refer to Lines 137-139 and 186-188] To measure few-shot difficulty, we used the same 20% split of $\mathcal{D}_N$ as used in SSL pre-training, but with label information. This is done to ensure that the remaining 80% portion used for evaluation of few-shot difficulty matches that used for evaluation of SL/SSL methods, for consistent analysis. We note that the 20/80 split is not a "class-split"; rather we perform a random split of the entire dataset regardless of class. Therefore, both partitions, for pre-training and fine-tuning/evaluation, respectively, contain all classes. We will clarify our partitioning scheme in the revision.
>
> -   [Please refer to Appendix O.1 in the revised manuscript] However, we agree with your concerns about the variance of few-shot difficulty according to the dataset split. Therefore, we analyzed the robustness of our few-shot difficulty measure w.r.t. to data partitioning. We calculate the few-shot difficulty of each dataset based on three distinct splits from different random seeds. The result shows that the ranking of few-shot difficulty between datasets do not change even if dataset splits are changed.
>
> -   [Please refer to Lines 191-192] We are fully aware and agree with the reviewer’s concern that few-shot evaluation is affected by the construction of the "5-way $k$-shot classification task." That is why we obtain the average performance across 600 randomly sampled 5-way $k$-shot episodes for all few-shot evaluations throughout the paper, following previous literature [1,2]. The same protocol is used when measuring few-shot difficulty.
>
> &nbsp;
> ### W2. Deeper backbone.
>
> [Please refer to Appendix O.2 in the revised manuscript] We added the results of ResNet50. We find that the observations in our paper also hold for ResNet50:
> -   SSL with strong augmentation can achieve higher performance than SL on datasets with small similarity.
> -   SSL performance gain over SL becomes greater at smaller domain similarity or lower few-shot difficulty.
> -   SL and SSL can synergize when they have similar performances, and two-stage training schemes improve the single-stage counterparts.
>
> &nbsp;
>
> [1] Phoo, Cheng Perng, and Bharath Hariharan. "Self-training for few-shot transfer across extreme task differences." International Conference on Learning Representations. 2021.
> [2] Guo, Yunhui, et al. "A broader study of cross-domain few-shot learning." European conference on computer vision. Springer, Cham, 2020.

---

> ### Author Response · Authors · 2022-08-09
> **Gentle Reminder**
>
> Dear Reviewer Ger5,
>
> We sincerely appreciate your valuable comments for improving our work. We have put in efforts in the revision (e.g., few-shot difficulty quantification and deeper backbone) to enhance our work based on your comments.
>
> We would like to send you a gentle reminder that the rolling discussion is now at the end of its period. We hope we have addressed your concerns. Please let us know if there is anything else you would like for us to address.
>
> Thanks,
> Authors

---

### Author Response · Authors · 2022-08-02
**General Response**

Dear reviewers and meta reviewers,

We appreciate all of the thoughtful comments including 1) the paper analyzes CD-FSL pre-training schemes and organizes the experimental presentation in a systematic way (**Reviewer Ger5**); 2) the paper reveals observations w.r.t. the domain similarity and few-shot difficulty through extensive experiments (**Reviewer eHxj**); 3) the paper is well-written and structured with an interesting experimental design that sheds light upon several points in CD-FSL (**Reviewer jPm7**).

We conducted additional experiments and analysis and attached the revisions to **the end of the supplementary file** (Appendix O), taking the reviewers’ comments into deep consideration. Due to the size of tables, we kindly refer reviewers to the revised appendix.

We summarize the main revisions as follows:
- Add results on the few-shot difficulty measure using different random seeds for data splits in Appendix O.1 (**W1** by **Reviewer Ger5**)
- Add CD-FSL performance of all main results using a deeper backbone, ResNet50, in Appendix O.2 (**W2** by **Reviewer Ger5**)
- Clarify line 224 (**W2** by **Reviewer eHxj**)
- Add t-SNE visualizations on trained representations and analyze causes for the difference between SL and SSL pre-training in Appendix O.3 (**W3** by **Reviewer jPm7**)
- Add CD-FSL performance of all main results under 20-shot and 50-shot evaluations, in Appendix O.4 (**Q2** by **Reviewer jPm7**)

If you have any further questions or suggestions, please share your comments on OpenReview. We will promptly address all the raised concerns in accordance with the reviewing policy.

---

### Author Response · Authors · 2022-08-08
**The end of the discussion phase is approaching**

Dear Reviewers,

Could you please go over our responses and the revision because we can have interactions with you only by this Tuesday (9th)? We have responded to your comments and faithfully reflected them in the revision. We sincerely thank you for your time and efforts in reviewing our paper, and your insightful and valuable comments.

Thanks, Authors.

---

> ### Comment · Area_Chair_TrLv · 2022-08-08
> **Review reminder: please respond to authors.**
>
> Dear Reviewers,
>
> Please remember to respond to the author rebuttal. Thank you.

---

### Meta-Review · Area_Chair_TrLv · 2022-08-20

**Recommendation:** Accept
**Confidence:** Certain

**Metareview:**

Authors present a comprehensive assessment of learning strategies for CD-FSL, including supervised learning, self-supervised learning (4 variants thereof),  semi-supervised learning (referred to as "mixed supervised"), single-stage and dual-stage training, and data augmentation strategies. Patterns are found among all these approaches by correlating results to source-target similarity by EMD, and measuring target task difficulty from SL.

Pros:
- [AC/R] A comprehensive study like this is currently missing from the CD-FSL literature. Some findings are less obvious than others and provide valuable information for the community.
- [AC/R] Extensive Experiments
- [AC/R] Well-written and easy to follow

Cons:
- [R] Assumption of unlabeled samples for the target domain is controversial. [AC] CD-FSL benchmark included unlabeled samples for study. While there may be applications where unlabeled data is not available, it does not diminish the utility of this setting.
- [R] Not clear if unlabeled and labeled data in target domain have overlapping classes. [AC] Assumption of benchmark is that there is overlapping classes, though acknowledge in practice this may not be the case. Probably an idea for a future benchmark rather than a critique of this work.
- [R] Some results are not surprising. Authors addressed this concern by pointing out that their results are in contradiction to prior findings, specifically around SSL.
- [AC/R] Only one distance measure was studied. Authors responded rationale for why EMD makes sense, but preferably they could have provided multiple measures and assessed how the multiple measures are similar and different, and how the findings may change.
- [R] Lack of visualization of embeddings. Authors have added visualizations to the appendix.

Overall reviews lean toward accept after revisions. Some negative comments from one reviewer are more critiques of prior published benchmark, not this particular work. Given this, AC feels assessment is mostly accept, though borderline. If authors could add additional distance measures and study how the different measures impact results, would improve quality and impact of paper. Recommend accept.

AC Rating: Borderline Accept


**Award:**

No

---

### Decision · Program_Chairs · 2022-09-14

Accept